# Decoding Neurodegeneration: A Comprehensive Review of Molecular Mechanisms, Genetic Influences, and Therapeutic Innovations

**DOI:** 10.3390/ijms241613006

**Published:** 2023-08-21

**Authors:** Victor Voicu, Calin Petre Tataru, Corneliu Toader, Razvan-Adrian Covache-Busuioc, Luca Andrei Glavan, Bogdan-Gabriel Bratu, Horia Petre Costin, Antonio Daniel Corlatescu, Alexandru Vlad Ciurea

**Affiliations:** 1Pharmacology, Toxicology and Clinical Psychopharmacology, “Carol Davila” University of Medicine and Pharmacy in Bucharest, 020021 Bucharest, Romania; victor.voicu@yahoo.com; 2Medical Section within the Romanian Academy, 010071 Bucharest, Romania; 3Department of Opthamology, “Carol Davila” University of Medicine and Pharmacy, 020021 Bucharest, Romania; 4Central Military Emergency Hospital “Dr. Carol Davila”, 010825 Bucharest, Romania; 5Department of Neurosurgery, “Carol Davila” University of Medicine and Pharmacy, 020021 Bucharest, Romania; razvan-adrian.covache-busuioc0720@stud.umfcd.ro (R.-A.C.-B.); luca-andrei.glavan0720@stud.umfcd.ro (L.A.G.); bogdan.bratu@stud.umfcd.ro (B.-G.B.); horia-petre.costin0720@stud.umfcd.ro (H.P.C.); antonio.corlatescu0920@stud.umfcd.ro (A.D.C.); prof.avciurea@gmail.com (A.V.C.); 6Department of Vascular Neurosurgery, National Institute of Neurology and Neurovascular Diseases, 077160 Bucharest, Romania; 7Neurosurgery Department, Sanador Clinical Hospital, 010991 Bucharest, Romania

**Keywords:** neurodegenerative disorders, molecular mechanisms, genetic mutations, frontotemporal dementia, tauopathies, protein-encoding gene

## Abstract

Neurodegenerative disorders often acquire due to genetic predispositions and genomic alterations after exposure to multiple risk factors. The most commonly found pathologies are variations of dementia, such as frontotemporal dementia and Lewy body dementia, as well as rare subtypes of cerebral and cerebellar atrophy-based syndromes. In an emerging era of biomedical advances, molecular–cellular studies offer an essential avenue for a thorough recognition of the underlying mechanisms and their possible implications in the patient’s symptomatology. This comprehensive review is focused on deciphering molecular mechanisms and the implications regarding those pathologies’ clinical advancement and provides an analytical overview of genetic mutations in the case of neurodegenerative disorders. With the help of well-developed modern genetic investigations, these clinically complex disturbances are highly understood nowadays, being an important step in establishing molecularly targeted therapies and implementing those approaches in the physician’s practice.

## 1. Frontotemporal Demetia (FTD)

Frontotemporal lobar degeneration (FTLD) syndromes exhibit a complex neuropathology characterized by heterogeneity. Two key forms are seen, known as FTLD-tau and TDP, distinguished by the presence of tau or TDP-43-positive inclusions, respectively [1]. Recently, FUS-positive inclusions have also been detected in some FTLD cases. Two rare neuropathological subtypes of FTLD exist. FTLD-UPS is characterized by inclusions positive for ubiquitin but negative for tau, TDP-43, and FUS. CHMP2B mutation cases tend to display this form, while FTLD-ni lacks discernable inclusions [2]. CHMP2B gene mutations were first identified in Danish kindred who suffered from frontotemporal dementia linked to chromosome 3 (FTD-3) [3]. This mutation alters the splice acceptor site for CHMP2B’s final exon, leading to the production of two novel transcripts known as CHMP2BIntron5 and CHMP2BDelta10. Furthermore, an autosomal dominant Belgian FTLD pedigree revealed another mutation known as CHMP2BQ165X, which causes premature stop codons, resulting in the protein lacking its final 49 amino acids, leading to its premature degradation [4].

Limited available data indicate that FTD occurs in approximately 11 cases per 100,000 individuals, with an incidence rate of 1.6 cases per 100,000 individuals. However, these figures appear to significantly increase from the fifth to seventh decades of life and likely underestimate actual prevalence due to misdiagnosis among older individuals. FTD accounts for 40% of dementia cases with early onset that have been confirmed through postmortem examination, although its onset typically begins between middle age and the ninth decade [5].

As serotonergic function is reduced in FTD, the use of serotonergic modulators is more apposite a priori, and modest behavioral benefit has been shown for trazodone and citalopram, though not for paroxetine [6]. Due to decreased serotonergic activity observed in FTD, serotonergic modulators appear more suitable as treatment options. Trazodone and citalopram have both demonstrated modest behavioral improvement; paroxetine did not achieve such benefits. If psychosis, aggression, or intrusive compulsions require management, then neuroleptic medications may be required, though care must be taken as individuals living with FTD can experience significant extrapyramidal side effects from even newer-generation antipsychotics. Therefore, in such instances, we suggest low-dose risperidone or quetiapine, with a strict clinical monitoring to avoid extrapyramidal side effects from antipsychotics [7]. 

Numerous potential strategies for treating FTD have been proposed. These include antitau antibodies and agents that stabilize microtubules, methods to increase the expression and release of progranulin, modulating autoimmunity and neuroinflammation (particularly applicable for GRN mutations), as well as using antisense oligonucleotides to silence toxic C9orf72 messenger RNA messengers—strategies which hold promise in targeting the root mechanisms while making advancements in FTD treatment [8]. 

GRN belongs to a family of growth factors with cysteine-rich polypeptides and can be found throughout various tissues. The GRN gene can be found 1.7 Mb upstream from MAPT’s microtubule-associated protein tau-encoding gene MAPT. GRN comprises 12 exons that code for its precursor glycoprotein of 68.5 kDa, divided into seven distinct granulins of 6 kDa each for secretion purposes by cells. GRN plays its role through cell signaling and signal transduction pathways [9].

Recent studies have demonstrated that mutations of the GRN gene are responsible for frontotemporal lobar degeneration, known as FTLD-U. This form is characterized by neuronal inclusions positive for ubiquitin but negative for tau. Mutations affecting GRN may lead to functional loss via an NMD process based on analysis of its cDNA both in brain cells and lymphoblastoid cells [10].

Frontotemporal dementia was initially clinically described by Arnold Pick in 1892. Later, Alois Alzheimer identified neuropathological lesions characteristic of Pick’s disease in 1911 [11]; these lesions, now known as Pick bodies, were later found in the 1960s to contain abnormal filaments made up of hyperphosphorylated microtubule-associated protein tau, and these neurofibrillary lesions closely resemble those described by Alzheimer in 1907, hence its naming after him [12].

In 1994, an autosomal dominant familial form of frontotemporal dementia with Parkinsonism was identified that was linked with chromosome 17q21.2. Missense mutations found in this form were thought to negatively impact how effectively tau protein interacts with microtubules; reduced interactions can be seen as partial loss of function, leading to destabilized microtubules that disrupt crucial cellular processes, such as rapid axonal transport [13]. 

Variants of FTD forms in other neurodegenerative disorders are as follows:-PiD (Pick’s disease) typically presents as behavioral variant frontotemporal dementia (bvFTD) or nonfluent variant primary progressive aphasia (nfvPPA), with motor deficits being rare. Histological characteristics of PiD include neuronal loss and swelling known as Pick cells; distinct large spherical neuronal cytoplasmic inclusions called Pick bodies may also be observed in some individuals with the disorder [14].-Progressive supranuclear palsy (PSP) typically presents as a movement disturbance characterized by early postural instability, axial rigidity, bradykinesia, and ophthalmoplegia. Cognitive impairment may be mild; however, some cases with PSP pathology show dementia similar to bvFTD or nfvPPA. PSP pathology also involves degeneration in multiple subcortical regions, including the striatum, globus pallidus, subthalamic nucleus, midbrain, tectum/tegmentum, substantia nigra, basis, pontis, cerebellar dentate nucleus, and cerebellar peduncles [15].-Patients diagnosed with corticobasal degeneration (CBD) often display corticobasal syndrome (CBS), characterized by bradykinesia, rigidity, dystonia, apraxia, cortical sensory signs, alien limb phenomenon, and bradykinesia [16]. -They may also show features of FTD (bvFTD or nfvPPA), displaying depigmentation in the substantia nigra, atrophy of globus pallidus, as well as focal and asymmetric cerebral cortical atrophy—histopathological features that overlap between PSP and CBD such as tau-immunoreactive glial cells as well as NCI histopathological features of both syndromes [17].-FTD caused by mutations of the MAPT gene is an autosomal dominant form linked to chromosome 17 (FTDP-17T) that accounts for roughly 10% of familial FTD cases. MAPT pathogenic mutations include missense or deletions in exons 1 and 9–13 or mutations after exon 10 that mainly manifest themselves through behavior changes, personality shifts, cognitive dysfunction, and atypical Parkinsonism, typically seen through behavior and personality alterations, cognitive deficits, and Parkinsonian-like symptoms as well as neuropathology featuring hyperphosphorylated tau deposits within gray and white matter structures [18].-Mutations to the VCP gene (valosin-containing protein) cause a rare familial syndrome known as inclusion body myopathy, Paget’s disease of bone, and FTD with variable penetrance (IBMFD). VCP belongs to the AAA-ATPase gene superfamily and serves as a molecular chaperone involved in various cellular activities that are either directly or indirectly controlled by UPS (ubiquitin–proteasome system) [19].

This histological pattern has been observed frequently among cases associated with tau mutations in various exons, including exons 9 (K257T, L266V, and G272V), 10 and 11 (L315R and S320F), 12 (E342V K369I), and 13 (G389R). Notably, in cases of FTDP-17 linked to I260V mutation in exon 9, brain scans did not reveal typical pathological tau species such as Pick-like bodies and neurofibrillary tangles, whereas S352L mutation also did not lead to the deposition of insoluble tau adopting pathological forms [20].

Patients who carry mutations in exon 10 or intron 10 typically display neuronal and glial tau pathology with ribbon-like filaments primarily composed of 4R tau. On the other hand, missense mutations outside exon 10 tend to present selective neuronal pathology that includes all six isoforms deposited as either paired helical filaments (PHFs) or straight filaments (SFs) [21].

Studies of FTD linked to chromosome 17 have been complicated by genetic heterogeneity. Some families that clearly link with 17q21 have been found not positive for MAPT mutations despite extensive analysis of their coding regions. Those families without tau pathology might suggest the presence of another defective gene at 17q21; however, MAPT mutations or complex forms, such as chromosomal rearrangements, cannot be entirely excluded as causes. Investigating the disease mechanism in these tau-negative FTD families is of vital importance, given that most FTD patients do not exhibit tau pathology according to immunohistochemistry tests. Investigating both tau-positive and tau-negative families should contribute to our knowledge about tau and related tauopathies, leading to more effective therapeutic approaches for this devastating disorder.

Targeting TDP-43 aggregates: One therapeutic strategy involves decreasing TDP-43 clearance or inhibiting their formation to help remove accumulations of toxic aggregates that accumulate soluble TDP-43 or toxic aggregates [22].

Targeting specific mutants: For specific mutations such as C9orf72, methods designed to decrease its transcript have shown promising results. Antisense oligonucleotides (ASOs) have been extensively researched and have proven successful at decreasing this mutated transcript. 

FTD-FUS (FTD fused in sarcoma) and therapeutic strategies: Studies have demonstrated that treating cell cultures with methylation inhibitors may help decrease cytoplasmic mislocalization and aggregates associated with FUS mutants in FTD cases. 

Mutations associated with FTD-UPS (ubiquitin–proteasome system): For therapeutic intervention, silencing the pathologically involved genes such as mutant CHMP2B by administering siRNA treatments has been observed to reverse cellular pathology in patient fibroblasts. FTD has several pathological conditions associated with it, yet no definitive mapping exists between these conditions and particular clinical presentations. One underlying condition often associated with FTD is “frontal lobe degeneration of the non-Alzheimer type,” or dementia lacking distinctive histopathology (DLDH), which helps distinguish its clinical presentation from any potential associated pathologies [23].

Human-induced pluripotent stem cells (iPSCs) are extensively employed in studies related to neurological and neurodegenerative conditions (Table 1). These iPSCs originate from specialized somatic cells, commonly from fibroblasts or peripheral blood mononuclear cells, through the heightened expression of the reprogramming agents Oct4, Klf4, Sox2, and c-Myc [24].

A distinct study noted marked tau pathology in N279K iPSC-derived neurons. Here, the neurons displayed shifts in the 4R:3R tau isoform balance, showcasing increased 4R tau levels and augmented tau fragmentation as early as 28 days after the maturation process of iPSC-derived neural progenitor cells (NPCs) [25].

In an initial analysis of patient-sourced iPSCs possessing the 10 + 16 mutation, a significant surge in 4R tau expression was documented during neuronal development. This led to a heightened 4R:3R tau isoform proportion in these cells [26].

iPSCs from patients with the N279K mutation were differentiated into NPCs. Subsequently, CRISPR/Cas9 genome editing was utilized to produce isogenic control cells. These patient-derived and isogenic control cells underwent differentiation into astrocytes by amplifying the expression of Sox10 and introducing pro-astrocytic growth and neurotrophic factors, including ciliary neurotrophic factor [27].

## 2. Spinocerebellar Ataxias (SCA)

Autosomal dominant spinocerebellar ataxias (SCAs, or ADCAs) comprise an expansive group of inherited ataxias with symptoms usually appearing between 30 and 50 years of age, although specific subtypes of SCA can appear earlier or even after 60 years [28].

Recent epidemiological evidence indicates that SCAs may be more prevalent than previously estimated. Prevalence rates exceeding 5–7 in 100,000 have been documented across various geographical regions; Japan, in particular, has reported 18.5 cases per 100,000 when considering dominant, recessive, and sporadic SCAs plus familial spastic paraplegia [29]. These numbers are comparable with other less frequent motor neurodegenerative conditions like Huntington’s disease or motor neuron diseases [30].

At present, over 30 SCA genes or loci have been identified, each one contributing differently to spinocerebellar neurodegeneration. Polyglutamine expansions in specific genes (SCAs 1, 2, 3, 6, 7, 17, and DRPLA) cause abnormally long polyQ tracts in encoded proteins. Noncoding expansions contribute to SCA 10 and 12 cases. Conventional mutations affect genes encoding cytoskeletal proteins (bIII spectrin in SCA5), voltage-gated potassium channel Kv3.3 in SCA13), protein kinases (tau Tubulin Kinase 2, SCA11; protein Kinase C Gamma 14), intracellular calcium channels (Inositol 1,4,5-triphosphate receptor 1), and fibroblast growth factors (FGF14 and ATPases AFG3L2 in SCA27 and 28) [31] (Table 2).

Dysregulation of transcription and gene expression is seen in SCA1, where ataxin 1 interacts with various transcription factors involved in transcriptional regulation, such as Lanp/Anp32a, PQBP-1, Silencing Mediator of Retinoid and Thyroid Hormone Receptors (SMRT), Boat, Gfi-1/senseless, Capicua, and Tip60. Mutant forms of ataxin 1 disrupt these transcriptional regulators’ activity by disrupting gene expression levels, leading to changes in Wnt-receptor signaling and Retinoic Acid/Thyroid Hormone signaling, as well as nucleic acid binding [32].

Other SCAs such as SCA2, SCA3, SCA7, SCA17, and DRPLA, as well as atrophin 1 gene product, directly participate in transcription as components of transcriptional regulatory complexes. For example, ataxins 2, 3, and 7 interact with basal transcription factor TATA-binding protein while atrophin 1 interacts with various transcriptional regulators—such interactions interfere with CREB-dependent transcription, which has an effect on gene expression [33].

The balance between protein acetylation and deacetylation, essential to optimal cell functioning, can be disturbed when mutant proteins contain expanded polyglutamine repeats that cause pathogenesis of neurodegenerative diseases. Restoring this equilibrium through a genetic or pharmacological adjustment in histone deacetylases (HDACs) potentially offers therapeutic strategies using HDAC inhibitors against neurodegeneration [34].

Alterations of synaptic neurotransmission play an important part in the neurodegenerative mechanisms underlying SCAs. Motor dysfunction precedes neuronal death in SCA1 transgenic mice, leading to Purkinje cell dysfunction that compromises Purkinje cell functions as well as changes to synaptic plasticity [35].

SCA8 mice share similar symptoms to human patients of this disease, specifically a loss of GABAergic inhibition in the cerebellum and intranuclear inclusions with expanded polyglutamine content in Purkinje cells and other neurons [36]. This finding could provide one explanation for SCA8’s lack of disease penetrance; it has been associated with large CTG repeat expansion in an antisense RNA for KLHL1 gene antisense RNA, as well as neuropathological analysis showing degeneration of Purkinje cells, inferior olivary neurons, and nigral neurons [37].

Alterations of calcium homeostasis play an integral part in SCAs. For instance, SCA6 is caused by polyglutamine expansions in the CACNA1A gene that encodes the alpha (2.1) subunit of CaV2.1 voltage-dependent P/Q-type calcium channel [38], while mutations of the protein kinase C gamma (PKC gamma) gene affect the C1 domain, essential for translocation and regulation of PKC gamma kinase activity [39]. Mutations in the PRKCG gene, which codes PKC gamma, result in the dysfunction of calcium homeostasis [40]. Finally, SCA15 causes deletions or missense mutations within the ITPR1 gene that are involved with intracellular calcium release from the endoplasmic reticulum (Table 2).

These changes in neurotransmission and calcium homeostasis play a key role in the pathogenesis of SCAs, providing therapeutic targets for intervention.

Mitochondrial Stress and Apoptosis: Polyglutamine-expanded cellular death of cerebellar neurons by polyglutamine-expanded containing proteins is preceded by the recruitment of caspases into polyQ aggregates. This is followed by the activation of caspases 3 and 9 and of mitochondrial apoptotic pathways mediated by members of the Bcl-2 family, such as Bax and Bcl-x(L). Both factors are known key components of neuronal apoptosis by regulating the mitochondrial release of cytochrome-c and Smac/DIABLO [41]. 

A method that has proven effective in diminishing the expression of mutant ATXN1 in vivo involves introducing AAV vectors that carry an shRNA targeting ATXN1 to the cerebellum [42].

This results in a significant reduction in mutant ATXN1 expression, leading to considerable enhancement in motor function and normalization of the structure of Purkinje cells in the transformed cells. Even though this SCA1 research utilized an shRNA targeting both the wild-type and mutant ATXN1, strategies focusing specifically on the mutant allele are being devised for numerous polyQ disorders, SCA3 included [43]. (Table 3)

In a sequence of research projects using a transgenic mouse model for SCA17, Xioa-Jiang Li and his team discovered indications suggesting that the polyQ expansion in TBP modifies its DNA-binding capacity and interaction with transcription regulators. This could potentially lead to the decreased expression of certain genes, such as HSPB1 and TrkA. Hence, for SCA7 and SCA17, the evidence suggests that polyQ expansion particularly affects these transcription regulators’ competency to manage the expression of certain key genes crucial for neuronal function, while not altering their regulation of the majority of other genes [44]. (Table 3)

## 3. Lewy Body Dementia (DLB)

Amyloid-beta protein has been implicated as an inducer of Lewy-type pathology, while mutations of alpha-synuclein can often result in cortical Lewy bodies as well as brainstem Lewy bodies; patients carrying these mutations frequently exhibit dementia symptoms. Other contributing factors of Lewy-type pathology include male gender, having late-onset Parkinson’s disease (PD), and carrying the CYP2D6*4 allele and specific alpha-synuclein haplotypes such as L478 and Rep1, among others. Mutations in the Parkin gene cause an early-onset autosomal recessive form of Parkinson’s disease characterized by changes to the substantia nigra without Lewy pathology development, suggesting that related metabolic pathways may influence vulnerable cell populations without leading to Lewy pathology development [45].

There may be an association between alpha-synuclein aggregation and Parkin mutations and proteasomal dysfunction and cell death pathways [46]; additionally, dementia found among DLB patients may be due to Alzheimer-type pathologies. Alpha-synuclein interacts with amyloid-beta protein through beta-amyloid’s ability to increase fibrillization and aggregation, leading to DLB pathology characterized by both Alzheimer-type pathology as well as alpha-synuclein pathology. Notably, alpha-synuclein aggregates may contribute to Parkinsonian features while amyloid-beta protein aggregates have been associated with Alzheimer’s disease (AD); hyperphosphorylated tau abnormalities may contribute to frontotemporal lobar degeneration [47]. 

Synucleinopathies are neurodegenerative conditions characterized by an accumulation of aggregated forms of the protein a-synuclein (a-syn) within various brain cells. Synucleinopathies, often associated with aging, are becoming increasingly prevalent due to extended life expectancy. Of all neurodegenerative conditions resulting in dementia, synucleinopathies rank second only to AD [48]. Most synucleinopathies fall under the category of Lewy body diseases (LBD), as they involve the build-up of aggregated a-syn in Lewy bodies within vulnerable neurons and Lewy neurites on neuronal processes. Parkinson’s disease, Parkinson’s disease dementia (PDD), and DLB are three of the more recognizable forms of Lewy body dementia; there may also be fewer common conditions [49]. 

Discovering a-syn as an essential component of LBD was made possible through findings of mutations of the SNCA gene (which encodes for a-syn) in familial forms of Parkinson’s disease and subsequent identification as one of the major components of Lewy bodies. Studies have demonstrated a strong link between mutations of SNCA and DLB occurrence sporadically. This association can be anticipated given that Lewy bodies contain a-syn, which has been implicated as playing an essential part in the pathophysiology of DLB, PD, and PDD. Unsurprisingly, research indicates a correlation between certain regions of the SNCA gene and different phenotypes of Parkinson’s disease and DLB; specifically, the 3′ region was linked with Parkinsonism while the 5′ region was linked with DLB pathology [50]. 

This may have an impact on the gene expression as well as brain distribution of Lewy body pathology. DLB has been linked with several genes, including SNCA, LRRK2, PSEN1, PSEN2, APP, SNCB MAPT SCARB2 GBA, and APOE. Noteworthy is the possibility that rare variants in AD-related genes (PSEN1, PSEN2, and APP) found in dementia cases could be misdiagnosed due to inadequate neuropathological assessment. Lewy body pathology is a relatively common feature of AD and may contribute to disease phenotype, leading to DLB. Recent genome-wide association studies have confirmed previously reported associations (APOE, SNCA, and GBA), and identified CNTN1 as an additional likely locus, providing an unbiased investigation of the genetics behind DLB [51]. 

APOE e4 allele and glucocerebrosidase (GBA) have emerged as two of the strongest genetic risk factors for DLB. APOE e4 allele is associated with an increased risk of DLB and is frequently found among individuals who exhibit mixed DLB-AD pathology, but is also overrepresented among cases of pure DLB and Parkinson’s disease dementia. Studies have established a correlation between APOE e4 and more severe Lewy body pathology, particularly among individuals with lower AD pathology [52]. 

GBA mutations are significantly more frequent among DLB patients compared to individuals without this condition. GBA mutations have been associated with early age of onset, higher disease severity, and faster disease progression in DLB cases. Similar to APOE, GBA may play an integral role in the formation and spread of Lewy body pathology, although its precise mechanisms have yet to be identified. Furthermore, an association was recently observed between DLB and SCARB2, a gene linked with Parkinson’s disease; this highlights lysosomal pathways as possible mediators of DLB development [53].

Studies conducted on DLB cases confirmed pathologically have revealed an association between GBA1 mutations and the condition. Initial investigations identified GBA1 mutations in an impressive percentage of DLB cases ranging from 3.5% to 28% depending on the specific research study conducted [54]. Subsequent multisite studies demonstrated GBA1 mutations in approximately 7.6% of DLB patients and 3.6% of individuals with both Lewy body disease and Alzheimer’s neuropathology [55]. These findings demonstrate that GBA1 mutations may be an influential risk factor for DLB and may impact disease development and progression. Furthermore, in a separate clinical study focused on Parkinson’s disease, GBA1 mutations had an enormous effect on its phenotype; specifically, the development of dementia symptoms in patients [56].

GBA1 mutations have been associated with earlier disease onset and death ages in DLB compared to noncarriers. DLB patients carrying GBA1 mutations generally experience disease onset approximately five years earlier compared to noncarriers, their median ages of disease onset being 63.5 and 68.9, respectively. The duration between diagnosis and death in cases carrying GBA1 mutations remains similar to noncarriers. Furthermore, they may demonstrate higher scores on the Hoehn Yahr scale and Unified Parkinson’s Disease Rating Scale than noncarriers, while E326K variant but not T369M has been associated with DLB and Parkinson’s disease dementia [57].

GBA1-associated Parkinsonism remains incompletely understood. Emerging evidence reveals that impaired lysosomal function, caused by deficient or mutant glucocerebrosidase enzyme, may impede alpha-synuclein degradation, an integral protein involved in DLB pathology [58]. 

Studies conducted using murine models and human neuronal cells demonstrate that dysfunctional glucocerebrosidase leads to accumulation and aggregation of alpha-synuclein, which, in turn, impairs trafficking and lysosomal activity of glucocerebrosidase activity, further worsening disease progression. Furthermore, autophagy disruption may contribute to its pathological mechanisms as part of DLB progression [59].

Recent biochemical studies have demonstrated similar levels of Ab40 and Ab42 in plaques between classic DLB and DLB with rapid progression (rpDLB), as well as abnormal solubility/aggregation of a-synuclein aggregates and increased binding to membranes of b-amyloid proteins in frontal cortex areas of both DLB and rpDLB cases [60].

Protein expression levels have been found to decrease significantly in DLB, with levels of NDUFA7, NDUFA10, NDUFB8, SDHB, UQCRC2, MTCO1, ATP5A, and ATP50 all showing significant reduction. No significant variations exist between rpDLB and DLB with respect to protein expression, with the exception of NDUFA7, which exhibits reduced expression even when normalized against VDAC expression levels, suggesting that rapid progression may have less of an impact on mitochondria compared to DLB [59].

Even though gene and protein expression vary significantly between DLB and rpDLB patients, the mitochondrial enzymatic activity of complexes I, II, III, and IV was significantly reduced in frontal cortex area 8 in both conditions, suggesting altered mitochondrial function as one major contributor to DLB and rpDLB pathogenesis in the frontal cortex [59].

Compared to DLB, three genes associated with purine metabolism (ENTPD2, NME3, and PRUNE) are significantly upregulated in rpDLB [59].

Additionally, reduced expression of NPM1 in the frontal cortex of DLB may indicate nucleolar stress linked to altered ribosomal biogenesis and protein expression of several transcription initiation factors at the ribosome, more so for rpDLB than DLB [61], while expression levels for elongation factors eEF1A and eEF2 were preserved across both types [62].

Notably, no significant variations were observed between DLB and controls in terms of gene expression for various cytokines and inflammatory mediators [63].

Genetic studies have identified variations in genes linked to other neurodegenerative disorders, including Parkinson’s disease (SNCA) and AD (APP, PSEN1, and PSEN2) among unrelated and sporadic DLB patients [64].

DLB was only recently recognized as a distinct disease entity, and research in molecular genetics of DLB lags behind that of Alzheimer’s and Parkinson’s diseases.

Rivastigmine, administered orally up to 12 mg/day, was the first to undergo a comprehensive evaluation in a large-scale (n = 120) randomized, double-blind, placebo-controlled international trial [65].

Remarkably, 63% of those treated with rivastigmine exhibited a 30% or greater enhancement on a digital cognitive assessment, contrasting with the placebo group, where only 30% exhibited improvement after 23 weeks. Rivastigmine-treated patients were observed to be less anxious and experienced fewer hallucinations. A comparable trend was detected with galantamine, given orally up to 24 mg/day, in a smaller (n = 50) open-label multi-center investigation on DLB patients. Here, a notable improvement from the starting point was seen on the Clinician’s Global Impression of Change (CGIC) scale (an increase of +0.5 out of 7 points; *p* = 0.01) after 24 weeks [66] Table 4.

Research using animal models for Parkinsonism and other neurodegenerative conditions has pointed towards excessive glutaminergic activity at cortical synapses. Memantine, an N-methyl d-aspartate (NMDA) receptor antagonist, benefits dementia patients by mitigating the harmful impacts of glutamate in their brains [67].

During the mild to intermediate stages of DLB, cholinesterase inhibitors such as rivastigmine, galantamine, and donepezil typically succeed in diminishing the frequency and intensity of hallucinations and delusions. Parkinsonian symptoms in DLB tend to be alleviated by the dopamine precursor levodopa, but its dosage must be moderated to prevent exacerbating visual hallucinations or the onset of unrest or excessive daytime drowsiness [68] Table 4.

Zonisamide, an anticonvulsant medication, obstructs sodium and T-type calcium channels and curtails the release of glutamate and carbonic anhydrase activity. It has secured approval in Japan for PD treatment. Its prevalent side effects involve weight reduction and diminished appetite, although these adverse reactions are infrequent [69].
ijms-24-13006-t004_Table 4Table 4Overview of targeted molecular therapies in Lewy body dementia.Treatment/DrugKey Details and OutcomesCitationRivastigmine-Administered orally up to 12 mg/day.-63% of treated individuals showed a 30% or greater cognitive improvement vs. 30% in the placebo group after 23 weeks.-Reduced anxiety and hallucinations [65,66]Galantamine-Administered orally up to 24 mg/day.-Notable improvement on the CGIC scale after 24 weeks.[66,68]Memantine-NMDA-receptor antagonist.-Benefits dementia patients by reducing harmful impacts of glutamate.[67]Levodopa-Alleviates Parkinsonian symptoms in DLB.-Dosage moderation is vital to prevent increased hallucinations or unrest.[68]Zonisamide-Anticonvulsant.-Approved in Japan for PD treatment.-Side effects include weight loss and reduced appetite, though infrequent.[69]

## 4. Friedreich’s Ataxia (FA)

Friedreich’s ataxia is an autosomal recessive trait. At first, only one locus on chromosome 9ql3 had been identified and mapped; however, more recently, FRDA2 has been proposed for certain families unrelated to chromosome 9 [70].

Friedreich’s ataxia affects approximately 2 to 4 cases per 100,000 individuals among Caucasians [71].

Nicolaus Friedreich, a German doctor, first described this condition between 1863 and 1877 in various reports. Two females and seven males from three sibships showed symptoms such as ataxia, dysarthria, sensory loss, muscle weakness, scoliosis, foot deformity, and cardiac symptoms. Onset typically occurred around puberty.

Friedreich’s ataxia patients most commonly exhibit an expansion of a GAA trinucleotide repeat within an Alu element within the first intron of the frataxin gene. Roughly 96% are homozygous for this GAA expansion while 4% possess both expanded allele and point mutation in their genetic coding sequence. Many missense mutations of frataxin occur in its carboxy-terminal half, an important functional region. Such missense mutations include L106S, D122Y, G130V, R165P/C, and L182F, which have been linked with milder clinical presentations among individuals heterozygous for such mutations [72].

The FRDA gene spans 80 kb of genomic DNA and comprises seven exons. Exons 1–5a encode a major transcript, 1.3 kb mRNA, which translates to frataxin protein with a 210-amino-acid sequence. Frataxin mRNA expression has been detected most prominently in the spinal cord and heart tissue as well as liver, skeletal muscle, and pancreas tissues; its distribution correlates with pathological features observed among those suffering from this condition [73].

Friedreich’s ataxia is an autosomal recessive disorder with variable symptoms, unlike many other recessive traits. Deviant variants have been mapped back to one locus on chromosome 9, including late-onset Friedreich’s ataxia (LOFA), with symptoms beginning 25 years after diagnosis; Friedreich’s ataxia with retained reflexes (FARR); and the Acadian form of Friedreich ataxia, which progresses more slowly without cardiomyopathy or diabetes mellitus [74].

Friedreich’s ataxia can be diagnosed through analysis of its GAA repeat. Molecular testing has uncovered unexpected presentations of Friedreich’s ataxia, such as pure sensory ataxia, spastic paraparesis, and chorea [75]. Although Geoffrey et al. proposed highly specific criteria for diagnosis, molecular testing may help avoid underdiagnosis in cases with very late-onset symptoms [76].

There have been various medications developed to alleviate oxidative stress caused by intracellular iron imbalance, which is either the result of frataxin deficiency directly or secondary effects thereof. Iron chelators such as desferoxamine and antioxidants like ascorbic acid or coenzyme Q10 analogs may be effective at decreasing iron overload in mitochondria; however, desferrioxamine only chelates iron from extracellular fluid and cytosol sources, not necessarily those in mitochondria. Therefore, its use should be limited strictly to controlled therapeutic trials only [77].

Studies have revealed an increase in urinary 8-hydroxy-2′-deoxyguanosine (80H2′dG), an indicator of DNA damage, in those suffering from Friedreich’s ataxia. This finding indicates that reactive oxygen species production could play a part in its pathogenesis. Treatment with antioxidant idebenone produced a significant drop in normalized urinary concentrations of 80H2′dG after just one dose in one small group of patients; further longitudinal research is needed to establish its efficacy as a biological marker and therapeutic benefits of other antioxidants or similar agents against this form of neurodegeneration [78].

Friedreich’s ataxia most often strikes young individuals of both genders. A systematic study with 115 patients from 90 families determined the mean age of onset as 10.52 ± 7.4 years, and death occurred 37.54 ± 14.35 years after initial symptoms appeared. Harding et al. reported these findings in their clinical and genetic investigation of 90 families; additionally, they investigated early diagnostic criteria as well as intrafamilial clustering of clinical features [79].

Not all patients of Friedreich’s ataxia possess homozygous GAA expansions in their frataxin gene. Two to four percent are compound heterozygotes containing one GAA expansion with either a point mutation or deletion on another allele; these individuals may exhibit some unusual symptoms, including less dysarthria than with homozygotes but a greater tendency toward optic pallor. Cossee et al. provided these data when investigating point mutations and clinical presentation among Friedreich’s ataxia patients [80].

Frataxin is a highly conserved protein essential for maintaining iron homeostasis and mitigating intracellular oxidative stress by modulating levels of reactive oxygen species (ROS). Cells lacking frataxin experience mitochondrial iron accumulation, an excess of reactive oxygen species, impaired antioxidant enzyme activity, and an increase in their susceptibility to oxidative stress. Marmolino et al. conducted research that demonstrated how an azelaoyl PAF PPAR-g agonist, known as APAF (azelaoyl PAF), could increase expression of frataxin mRNA and protein in primary fibroblasts from both healthy individuals and Friedreich’s ataxia patients; similar effects were also observed with neuroblastoma cell line called SKNBE, suggesting that APAF might regulate the expression of the FXN gene in tissues relevant to disease [81].

Past studies have investigated the efficacy and safety of BML-210 as a histone deacetylase inhibitor compound to treat Friedreich’s ataxia in mice carrying GAA230 repeats, using human lymphoblastoid cells and 3-nitropropionic acid as examples of such compounds with potential. Their results suggested promise as potential therapies; however, some have demonstrated mitochondrial toxicity, so further investigation should take place to evaluate their safety and effectiveness for humans [82]. 

Based on previous observations, fibroblasts of patients with Friedreich’s ataxia (FRDA) exhibited decreased cytoskeletal organization and elevated levels of glutathione bound to cytoskeletal proteins. Furthermore, abnormal immunoreactivity for glutathionylated proteins such as GS-Pro (glutathione bound to proteins) was seen both in gray matter neurons as well as white matter cells and axons of patients, suggesting abnormal protein glutathionylation attributed to FRDA. These results suggest the presence of oxidative stress due to reduced frataxin expression; glutathionylated proteins serve as biomarkers of such effects of stress [83].

Studies involving mitochondria isolated from adipocyte-like cells that overexpress frataxin demonstrated an association between frataxin and mitochondrial calcium (Ca^2+^). Frataxin expression led to activation of the respiratory chain, increased membrane potential and production, elevated ATP production, as well as the uptake of mitochondrial Ca^2+^. This increase may have stimulated tricarboxylic acid cycle activity, indicating frataxin’s connection with energy conversion through mitochondria [84].

Frataxin-deficient dorsal root ganglia sensory neurons produced an increase in intracellular free Ca^2+^ following its reduction, leading to caspase activation, increased nitrosation, and activation of pro-apoptotic gene Bax activation—events observed in FRDA patients as fragmentation of alpha-fodrin, axonal degeneration, and eventual apoptosis of these neurons, as seen through altered calcium homeostasis and cell death. Furthermore, it was determined that depletion could be prevented via its BH4 domain of Bcl-xL protein, which plays an active role in both cell death as well as altered calcium homeostasis regulation [85].

Alterations of mitochondrial bioenergetics and Ca^2+^ homeostasis of FRDA neurons could have profound ramifications on axonal transport and distribution along axons and synapses, which is particularly noteworthy given the extensive axonal network. Studies on Drosophila larvae focusing on mitochondrial transport showed that decreased frataxin expression led to early decreases in membrane potential, defects in retrograde transport (primarily affecting retrograde direction), abnormal accumulation of mitochondria at synapses, and dying-back neuropathy. These findings suggest that mitochondria may contribute significantly to the impairment of axonal transport in FRDA patients [86]. 

## 5. Progressive Supranuclear Palsy (PSP)

PSP is an uncommon neurodegenerative disease with an estimated prevalence of about 7–10 individuals per 100,000, considered one of the more frequent atypical Parkinsonian syndromes and marked by motor, behavioral, and language abnormalities [87].

PSP displays clinical heterogeneity and has many different phenotypes, with Richardson’s syndrome (PSP-RS), also known as Steele–Richardson–Olszewski syndrome, being the classic example, first described in 1964 and first identified by its symptoms of vertical gaze palsy, pseudobulbar palsy, nuchal dystonia, and dementia [88].

Pathologically, PSP can be described by the presence of 4-repeat (4R) tau inclusions such as neurofibrillary tangles, neuropil threads, tufted astrocytes, and oligodendroglial coiled bodies, predominantly found in basal ganglia, diencephalon, and brainstem, with prominent involvement of globus pallidus subthalamic nucleus and substantia nigra. PSP is the most prevalent primary 4R-tauopathy, with abnormal accumulation beginning prior to the presymptomatic phase. Although most commonly seen sporadically, there have been familial forms as well as a few pedigrees showing characteristics similar to PSP, which suggest an autosomal dominant inheritance pattern [89].

Mutations in the microtubule-associated protein tau (MAPT) gene have been implicated in PSP, with approximately 0.6% to 14.3% of cases carrying MAPT mutations. Individuals diagnosed with PSP often carry multiple mutations that contribute to its development and worsen its disease phenotype [90].

One mutation, known as the S305S mutation of the tau gene, has been linked with mild cortical atrophy and extensive subcortical neurofibrillary tangles, consistent with the neuropathological diagnosis of PSP. Moreover, neuropathology is distinguished by a prevalence of subcortical neurofibrillary and glial tangles, neuropil threads composed mainly of hyperphosphorylated tau filaments, and thread-like neuropils that often appear. Pathological changes occur predominantly in the basal ganglia, brainstem, and cerebellum without significant amyloid deposition. Neuronal loss has been noted in both output nuclei of the basal ganglia as well as the focal frontal and temporal cortical regions; frontotemporal atrophy remains minimal while specific pathological features, including ballooned neurons and tufted astrocytes, can be observed within affected brain regions [91].

PSP neuropathology serves as an authoritative benchmark, offering insight into its underlying mechanisms and providing a definitive diagnosis.

Transactivation-responsive DNA-binding protein of Mr 43 kDa (TDP-43) is a nuclear protein that plays an integral part in transcriptional repression and alternative splicing. It was initially identified as one of the major contributors to abnormal protein accumulations found in the frontotemporal cortex and motor neurons of people diagnosed with frontotemporal lobar degeneration-U (FTLD-U) or amyotrophic lateral sclerosis (ALS), marked by both ubiquitin-positive and tau-negative inclusions [92].

TDP-43 accumulation was initially believed to be limited to FTLD-U and ALS, but subsequent studies have demonstrated abnormal accumulation in certain cases of other neurodegenerative diseases, including AD, Parkinson’s disease with and without dementia, DLB, etc.

Recent research demonstrated that a significant proportion (26% to be exact) of patients with PSP, typically classified as tauopathy, had abnormal accumulations of TDP-43 in the limbic system, suggesting that pathological TDP-43 accumulation can also occur within this form of tauopathy and could contribute to hippocampal sclerosis within this form of PSP cases [92].

Early studies involving TDP-43 immunohistochemistry and immunoblot analysis with nonspecific antibodies suggested that PSP cases do not exhibit abnormal accumulation of TDP-43 protein. However, more specific antibodies and biochemical analysis revealed alterations in TDP-43 pathways similar to those seen in FTLD-TDP and ALS patients [93].

However, previous studies of AD suggest that the presence of TDP-43 pathology is associated with later age of onset and death as well as diminished cognitive function [94].

Some cases of early-onset autosomal dominant PSP have been linked to mutations (G303V) in the tau gene that result in the overexpression of 4R isoform tau and its hyperphosphorylation, producing specific protein bands on analysis. Mutations in this gene are relatively uncommon but certain polymorphisms and haplotypes have been found associated with PSP, while individuals carrying mutations may exhibit atypical features of PSP [95].

Levodopa, combined with a peripheral decarboxylase inhibitor like carbidopa or benserazide, stands as the primary drug used for dopaminergic replacement therapy in PSP. Typically, the PSP–Parkinsonism (PSP-P) subtype shows a more pronounced response. Symptoms like bradykinesia, rigidity, and tremor in any PSP phenotype might respond comparably to those in PSP-P, but improvements in postural stability are less likely. Retrospective research suggests that 20–30% of pathologically confirmed PSP patients and 20–40% of clinically diagnosed PSP patients have observed positive effects from levodopa, either as a standalone or in tandem with another dopaminergic drug like amantadine [96] Table 5.

Despite anecdotal claims of marked improvement in motor functions on low doses of amitriptyline, data on pharmacological treatments for gait issues in PSP remain scanty. The ineffectiveness or lack of response of gait problems or freezing to levodopa and other dopaminergic drugs is a common observation. Amantadine, an NMDA-receptor antagonist, has shown inconsistent results [97] Table 5.

A retrospective analysis involving 147 clinically diagnosed patients indicated that carbidopa–levodopa was the most potent agent, enhancing Parkinsonism in 20% of the treated group. Dopamine agonists, amantadine, and selegiline did not yield favorable outcomes [98].

Fluctuations related to dosage in dopaminergic response have not been distinctly reported for PSP/CBS. Thus, there is no requirement for controlled-release or extended-release versions or for monoamine oxidase B inhibitors or catecholamine O-methyltransferase inhibitors in such contexts [99].

## 6. Corticobasal Degeneration (CBD)

Rebeiz, Kolodny, and Richardson first introduced CBD as an entity between 1967 and 1968. Three patients displayed symptoms including slow and awkward voluntary limb movement, tremor, dystonic posturing, stiffness, lack of dexterity, and the sensation of “numbness or deadness” in one or both affected limbs. Over time, these symptoms intensified and included gait disorder, limb rigidity, impaired position sense, and other sensory deficits. Pathological examination revealed an asymmetric frontoparietal cortical atrophy and neuronal loss with associated gliosis. Nissl substance was absent in affected neurons, thus giving rise to the term “achromatic.” Swollen eosinophilic and hyaline appearing pyramidal neurons were observed in the third and fifth cortical layers; there was significant loss of pigmented neurons from substantia nigra, variable involvement among subcortical neurons, as well as secondary degeneration of corticospinal tract [100].

CBD usually presents itself in middle to late adulthood, with the average age of onset being 63 years old [101]; however, cases have been reported as early as 40 years old and confirmed through pathology at 45 years. Both males and females can be affected, although some studies have noted a greater number of women [102].

Studies show that tau protein in CBD comes from transcripts of exon 10 on chromosome 17. Pathological tau displays abnormal phosphorylation and solubility properties as well as abnormal filamentous structures. Ballooned neurons, neurofibrillary tangles, neuropil threads, grains, glia, and neuronal inclusions all display tau immunoreactivity; neurofibrillary lesions showing tau immunoreactivity can often be found in locations like locus ceruleus, raphe nuclei, tegmental gray matter, or even the substantia nigra [103]. 

CBD, with its characteristic tau-immunoreactive astrocytic lesion that resembles AD, shares several traits with Alzheimer’s. These tau-positive astrocytic plaques are argyrophilic structures often observed in CBD, although they have also been identified in PSP. Unlike AD, however, these plaques in CBD originate from glial cells rather than being amyloidogenic in origin [103]. CBD is associated with specific tau-immunoreactive lesions in neurons and glial cell processes. Minimal pathological features for diagnosing CBD may include cortical and striatal tau-positive neuronal and glial lesions, particularly astrocytic plaques and thread-like structures in white matter; neuronal loss in specific cortical regions and the substantia nigra may also indicate its presence [104].

CBD and PSP share characteristics, as evidenced by identical H1 and H1/H1 tau haplotype statuses seen across both disorders. This shared genetic predisposition suggests that both disorders share a common genetic factor contributing to their development; however, these may still represent separate nosological entities caused by different etiological factors occurring simultaneously or extreme forms of the same disorder arising due to variations in genetic background or specific trigger agents [105].

Studies have uncovered decreased complex III activity and an increase in markers of lipid oxidative damage among PSP cybrid cell lines containing mitochondrial DNA from PSP patients [106], but contradictory reports regarding aconitase activity across both types of cell lines suggest that they do not always accurately reflect changes occurring in human brain tissue postmortem [107].

Patients presenting initially with primary progressive aphasia (PPA) or frontotemporal dementia may later develop movement disorders or symptoms characteristic of CBD, typically two to seven years after the initial manifestation occurred.

The treatment of Parkinsonism in CBD/CBS often involves levodopa, though it generally yields minimal effects. Some reports suggest a brief mild to moderate improvement [101] Table 6.

Other therapeutic approaches for Parkinsonism, encompassing other dopaminergic treatments, benzodiazepines, and anticholinergics, typically do not offer much assistance and can sometimes lead to side effects, like cognitive decline [98] Table 6.

Dystonia tends to respond best to precise botulinum toxin injections. In the context of CBS, botulinum toxin shots can be administered to alleviate pain, enhance hygiene, counteract secondary contractures, and occasionally bolster limb functionality during the early disease stages. Although some practitioners might experiment with oral medications like benzodiazepines, anticholinergic drugs, or muscle relaxants, they seldom prove effective. The principal line of treatment remains botulinum toxin injections [108] Table 6.

## 7. Wilson’s Disease

Wilson’s disease, also referred to as hepatolenticular degeneration, is an autosomal recessive genetic condition caused by mutations in the ATP7B gene. Reports of individuals suffering from symptoms including liver cirrhosis and neuronal degeneration date back as far as 1850; however, Kinnear Wilson first formalized this clinical entity in 1912 by identifying seven familial patients suffering from progressive degeneration of their lenticular nuclei and liver cirrhosis upon autopsy. Wilson speculated that there may have been toxic chemicals affecting liver cirrhosis, but copper’s role in Wilson’s disease pathogenesis was only recognized 35 years after his initial observations [110].

Scheinberg and Gitlin first described Wilson’s disease in 1952 after noting a deficiency of ceruloplasmin in affected individuals’ serum, a diagnostic test that remains important today. Wilson’s disease can occur anywhere worldwide and affects individuals aged 3 to 80; children and adolescents seem more prone than adults, due to differences in estrogen levels and iron metabolism [111].

Wilson’s disease affects approximately 1 out of every 30,000 individuals globally, with an approximate carrier rate of approximately 0.011 and a gene frequency of 0.56. Recent clinical studies have reassessed its prevalence [112]. 

Hepatocytes, the primary target for copper uptake and accumulation in the liver, possess a remarkable ability to assess intracellular copper status and control its excretion through bile [113]. This regulation is enabled by an ATPase encoded at Wilson’s disease locus that transports copper out of cells into trans-Golgi networks [114]. Hepatocytes express this enzyme widely. As copper levels in hepatocytes increase, ATPase moves from its original location within the trans-Golgi network to a vesicular compartment near the canalicular membrane. Copper accumulates within this vesicular compartment, leading to decreased cytoplasmic copper levels and prompting ATPase redistribution back towards the trans-Golgi network [115]. Ultimately, copper excretes into bile for excretion from cells, effectively maintaining intracellular copper homeostasis while simultaneously rapidly eliminating excess cytosolic copper levels [116] Figure 1.

Copper has no direct effect on the production or secretion rate of ceruloplasmin, an essential protein responsible for copper transport in blood plasma, yet failure to incorporate copper results in secreting non-functional forms that quickly degrade in plasma. Wilson ATPase plays an essential role in transporting copper to secretory pathways of hepatocytes; impaired movement into these compartments results in the disruption of ceruloplasmin production, leading to significant drops in serum ceruloplasmin levels due to Wilson’s disease [117].

Genetic studies conducted with yeast have provided insight into the role of atox1 protein as a copper chaperone for delivery to the secretory pathway. Atox1 interacts with Wilson ATPase in an atox1-dependent fashion, and mutations that conserve copper-binding domains (an amino terminus of Wilson ATPase) lead to decreased atox1 binding; these observations suggest that impaired delivery could be the source of Wilson’s disease in patients carrying such mutations, thus making atox1 an essential copper chaperone that plays critical roles in copper homeostasis regulation and homeostasis regulation [113]. 

Wilson’s disease (WD) patients do not all carry causative mutations. One such gene, located on 13q14.3, spans 80 kb of genomic space and contains 21 exons encoding ATP7B or Wilson ATPase, which transport copper via P-type ATPase transporters. This ATPase contains 1465 amino acids and is synthesized in the endoplasmic reticulum before localizing to the trans-Golgi network (TGN) of hepatocytes. However, its expression has also been detected in various other tissues including the brain, kidney, lung, and placenta. Over 500 mutations of the ATP7B gene have been identified, most being extremely rare [118]. Mutations include missense or nonsense single-nucleotide variants (60%), insertions/deletions (26%), and splice site mutations (9%). Compound heterozygotes are often observed among WD patients where multiple different mutations exist on both gene copies [119].

ATP7B gene mutations vary significantly across geographical regions. Some mutations, like H1069Q and R778L, are particularly prevalent among European and Asian populations, while most other reported mutations have an incidence rate below 10%. Hotspots for WD gene mutations in Europe can be found between exons 8 and 18. In India, however, mutations within exons 2–5 have been linked with severe phenotypes [119].

H1069Q mutation in exon 14 is the most frequently occurring type, occurring in 30–70% of WD patients. This mutation involves replacing histidine from the SEHPL motif in the N domain with glutamic acid, leading to misfolding and abnormal phosphorylation of the P domain, resulting in reduced ATP binding affinity of half the normal level; additionally, it shows decreased thermal stability as endoplasmic reticulum abnormally migrates toward TGN localization [120].

Modifier genes are known to exacerbate or alleviate phenotypes caused by other disease-causing genes. One such modifier, apolipoprotein E (ApoE), found on 19q13.2, has been strongly linked with Western dry eye disease. ApoE exists as three alleles: e2, e3, and e4. Of these alleles and genotypes, ApoEe3 and e3/3 are most often found together and play an essential role in lipid metabolism, while ApoEe4 is linked with neurodegenerative disorders like AD as well. Evidence shows that patients with the e3/3 genotype may delay their onset age due to the neuroprotective and antioxidative properties of ApoE3, while ApoEe4 has been associated with earlier-onset symptoms of WD [121].

Penicillamine was one of the first successful oral drugs designed to treat Wilson’s disease, providing many patients with lifesaving therapy. It has become widely prescribed, but nowadays, this can become an issue when physicians prescribe it while being unaware of safer pharmacological options. Penicillamine works by chelating copper from liver cells. Unfortunately, due to its toxicity, it also acts as a known teratogen, which may lead to neurological symptoms in unborn babies if administered as initial treatment in pregnant patients with neurological symptoms [122].

Among patients treated with penicillamine, 50–75% experience further neurological deterioration and 25–35% do not return to pre-penicillamine levels of functioning; one out of every four treated may suffer permanent additional disturbances and severe disability as a result of taking penicillamine due to copper mobilization from their livers leading to further elevation in brain copper levels. This worsening of symptoms could be attributed to its mobilization, leading to increases in brain copper levels as a result.

Trientine was first introduced as an alternative to penicillamine in 1982 as an option for patients who are sensitive to it, specifically those intolerant of it. Like penicillamine, trientine also acts as a chelator, inducing urinary excretion of copper; however, trientine is generally considered safer as its dose and administration method remain similar. Another advantage is that trientine does not trigger hypersensitivity reactions associated with penicillamine. Moreover, trientine has been extensively studied as an initial therapy instead of penicillamine. Nevertheless, trientine is generally seen as a possible alternative when administered instead of penicillamine for those intolerant of penicillamine treatment plans [123]. 

Tetrathiomolybdate (TM) is an experimental treatment for Wilson’s disease. When taken with food, TM works by binding copper and protein together and producing an imbalance that prevents food absorption as well as endogenously secreted copper from reaching your system, leading to reduced overall copper levels in your body. When taken between meals, TM absorbs into the bloodstream, where it forms complexes with blood copper and albumin, which makes this complexed copper non-available for uptake by cells, making this non-toxic [124].

Dysarthria is one of the primary neurologic manifestations of Wilson’s disease, affecting approximately 89% to 97% of patients who exhibit neurological involvement. Dysarthria typically takes on a mixed form characterized by spastic, ataxic, hypokinetic, dystonic, and spastic components, often congruent with dystonia as it displays dystonic qualities with strained or harsh voice qualities; conversely, those suffering from Parkinsonism may exhibit hypokinetic properties when speaking [125].

Dystonia is a characteristic of Wilson’s disease that is found in approximately 11–65% of patients, often manifesting itself through focal, segmental, multifocal, or generalized dystonia ranging in severity from mild to severe. One notable manifestation is “risus sardonicus”, an exaggerated and forced smile caused by dystonic facial muscles [126].

Wilsonian tremor can be observed in 22–25% to 55% of individuals diagnosed with neurologic Wilson’s disease. This tremor may occur at rest, when taking certain postures, or while performing actions. Parkinsonism, marked by bradykinesia, cogwheel rigidity, or imbalance, affects 19–62% of patients living with Wilson’s disease [125].

Wilson’s disease patients experience seizures at a rate that is ten times higher than in the general population; however, seizures rarely present themselves initially and have often been associated with initiating chelating therapy [127]. 

Kayser–Fleischer rings, copper deposits in the limbic region of the cornea, can be found in nearly 100% of those suffering from neurologic Wilson’s disease and 50% with presymptomatic and hepatic forms of it. Sunflower cataracts do not impair vision but must be detected with an ophthalmoscope; their visibility cannot be observed through naked-eye viewing alone [128].

Distinguishing Wilson’s disease from other common and rare neurologic conditions is crucial, given that Wilson’s is treatable. Young individuals experiencing movement issues should consider it as a possible diagnosis. Wilson’s disease can often be confused with essential tremor, young-onset Parkinson’s disease, and generalized dystonia, all common neurologic conditions that often present themselves similarly. Rare juvenile genetic extrapyramidal disorders, including Huntington’s disease, Hallervorden–Spatz’s disease, idiopathic torsion dystonia, chorea-acanthocytosis, and benign familial chorea can all present with symptoms similar to Wilson’s disease. Wilson’s disease should certainly be considered in any attempt at differential diagnosis; however, other potential causes must first be explored. Psychological abnormalities associated with Wilson’s disease could easily be misinterpreted as affective disorders, early schizophrenia, or drug abuse, requiring careful differential diagnosis to ensure appropriate diagnosis [128].

Dimercaptosuccinic acid (DMSA) is an alternative maintenance therapy for Wilson’s disease, with lower urinary copper excretion and more severe adverse effects than dimercaptopropanol. Na-DMPS (sodium dimercaptosuccinate) has low toxicity but may not be appropriate for advanced Wilson’s disease or critically ill patients; calcium disodium edetate (CaNa2-EDTA) also chelates copper less effectively compared with zinc or iron. Therefore, DMSA/NaDMPS is recommended as alternative maintenance therapy when D-penicillamine cannot be tolerated [129].

Zinc has long been recognized as an effective adjuvant therapy to decrease copper absorption in Wilson’s disease patients. Studies have demonstrated that oral zinc sulfate or zinc gluconate significantly increased urinary copper excretion, and clinical symptoms improved dramatically for those using zinc sulfate as part of maintenance therapy during follow-up. Zinc also has shown great efficacy with both preclinical patients as well as those using copper chelators; they even show promising clinical efficacy among asymptomatic and preclinical patients and those in maintenance after starting with copper chelators treatment [130]. 

Studies on Wilson’s disease using magnetic resonance imaging (MRI) have extensively explored its structural changes, which correspond with abnormal postsynaptic dopaminergic function as measured by [11C] raclopride PET/IBZM-SPECT scans [131]. Wilson’s disease patients typically do not respond well to levodopa treatment due to abnormal dopaminergic function, possibly explaining their subpar response to it. PET and SPECT studies have also indicated the presence of presynaptic dopaminergic damage comparable to that seen in Parkinson’s patients; this damage may contribute to occasional levodopa responsiveness. Such presynaptic damage could have an impactful influence on neurological symptoms in particular patients [132].

Wilson’s disease causes presynaptic damage that varies widely among its patients, leading to inconsistent reports of levodopa responsiveness and no correlation between extrapyramidal symptoms and MRI findings [133]. There is a notable relationship between extrapyramidal symptoms and olfactory dysfunction. Those suffering from marked deficits could potentially benefit from dopaminergic medications; however, clinical experience with dopamine-antagonistic medication has not proven very fruitful, further disproving any essential role played by this pathway in neurological-type Wilson’s disease patients [134].

An intriguing possibility is that therapy-induced increases in dopamine D2-receptor binding could signal functional recovery of the striatal dopaminergic system, possibly linked to olfactory function. Therefore, metabolically induced damage of nigrostriatal structures could contribute to Wilson’s disease as olfactory dysfunction is caused by metabolic damage to specific structures within basal ganglia that process odorous stimuli; further research is necessary in this area of dopaminergic function, structural changes, and deficits within Wilson’s disease [135].

Wilson’s disease stands out from its counterparts in that its usual progression pattern does not apply; rather, there appears to be an apparent incongruence [136]. Pathological studies have revealed no damage in the substantia nigra region commonly affected by Parkinson’s disease [137]; instead, research shows nigrostriatal dopaminergic damage occurs more commonly at nerve terminals rather than cell bodies, which corresponds to the functional nature of lesions found in Wilson’s disease [137].

Wilson’s disease usually results in lower total serum copper levels due to reduced ceruloplasmin levels; however, under certain circumstances, such as severe liver injury or acute hepatic failure due to Wilson’s disease, these may rise significantly due to copper released from damaged liver tissue stores and stored reserves [134].

Copper can be detoxified when it binds covalently to high-affinity copper-binding proteins that act as chaperones or endogenous chelators within cells, acting as chaperones or endogenous chelators within cells. Copper bound to ceruloplasmin in the bloodstream is non-toxic, while remaining serum copper, known as free copper, may become toxic through less-tight binding to proteins and being mobilized more readily, potentially leading to copper toxicity and contributing to reactive oxygen species formation. Additional research implicates acid sphingomyelinase and ceramide in liver cell death due to mitochondria being the prime target in liver cell death; more recent research has indicated acid sphingomyelinase and ceramide in liver cell death [138].

Wilson’s disease does not typically involve changes to either the olfactory bulb or piriform cortex; instead, their symptoms could be related to direct, metabolically damaged dopaminergic nerve terminals projecting to olfactory areas in the brain that project these scents [139].

Although chelator-induced neurologic worsening and free copper toxicity appear to have an apparent relationship in Wilson’s disease, further clinical and experimental data must be collected to fully establish this concept. Reacting to biochemical readings before clinical worsening could potentially benefit patients; however, certain limitations must also be considered [140].

Calculating non-ceruloplasmin-bound copper involves precise measurements of both copper and ceruloplasmin levels; however, inflammation or immunologic assays that detect nonfunctional protein precursors (e.g., apo-ceruloplasmin) can lead to overestimation of ceruloplasmin levels, leading to inaccurate values for non-ceruloplasmin-bound copper and making their calculations useless [141].

To address this challenge, direct measurement of the lower-affinity or non-covalently bound copper pool has been proposed as an approach to assess copper toxicity in Wilson’s disease more easily and reliably. Ultrafiltration or other techniques could be employed for this purpose. Measuring serum-free copper may offer a simpler and more reliable means of doing so [141].

Exchangeable copper (CuEXC) and its relative exchangeable copper (REC), both promising approaches that could offer valuable insights into copper dynamics and disease progression, are also promising approaches. But for these strategies to become truly useful clinical tools, further research and evidence must be collected and utilized effectively in clinical practice [142]. 

A model incorporating the R778L mutation was effectively differentiated into hepatocyte-like cells (HLCs), suggesting the successful creation of a Wilson’s disease model from hESCs without needing a patient sample. Notably, HLCs with the R778L mutation displayed a heightened susceptibility to excessive copper, making them more sensitive to copper-induced cytotoxicity than the standard HLCs. This newly developed model, which does not require a patient sample, offers a fresh avenue for testing the efficacy of drugs before their clinical use [143] Table 7.

## 8. Niemann–Pick Disease

Since the early 1980s, Niemann–Pick disease has been divided into two distinct entities. Acid sphingomyelinase deficiencies encompass types A and B, and Niemann–Pick disease type C includes types C and D. This illness results from deficiencies in either NPC1 or NPC2 transport proteins, leading to lipid storage diseases [144].

Acid sphingomyelinase-deficient Niemann–Pick disease, more commonly referred to as ASM-deficient NPD or ASM deficiency, is an autosomal recessive disorder caused by mutations of the SMPD1 gene. A deficiency of ASM lysosomal enzyme leads to progressive accumulation of sphingomyelin throughout body organs in all forms and neuronopathic forms, with secondary accumulations of other lipids as well [145].

History shows us that ASM deficiencies can generally be divided into two broad types, known as types A and B. Although intermediate forms have also been observed, ASM deficiency represents an entire continuum that ranges from severe neuronopathic forms such as type A, more prevalent among Ashkenazi Jews than other populations, to non-neuronopathic forms like type B, with estimated incidence rates estimated at one out of every 200,000 births in France and intermediate forms more frequent throughout central Europe [144]. 

Classical Niemann–Pick Disease Type A: Neonatal life usually begins normally, with symptoms like vomiting or diarrhea typically manifesting within months of life. Poor weight gain and an enlarged liver and spleen (hepatosplenomegaly) may also signal classical Niemann–Pick disease type A. Nearly 70% of cases may exhibit mild physical abnormalities and brownish pigmentation of their skin, with neurological signs typically appearing between 5 and 10 months of age, beginning with decreased muscle tone (hypotonia) before eventually progressing to progressive loss of acquired motor skills and interest in their surroundings. Nerve conduction typically slows, and macular cherry-red spots may develop at an advanced stage of the disease. Hypotonia then progresses to bilateral pyramidal signs, spasticity increases over time, deep tendon reflexes become absent, and seizures may occur but usually do not pose major threats. Death typically occurs between 1.5 and 3 years, although in some instances, symptoms may develop over an extended period of time [146].

Niemann–Pick Disease Type B: Niemann–Pick disease type B is a non-neuronopathic form, with variable degrees of systemic involvement and diagnosis typically in late infancy or childhood; it can sometimes also appear later in adulthood. Most patients living with type B will reach late adulthood, while some children may develop severe systemic disease, leading to their premature demise [147].

Niemann–Pick Disease Type C (NP-C) is an autosomal recessive atypical lysosomal lipid storage disorder caused by mutations of either of two genes: NPC1 (95% of cases) or NPC2. Both genetic variants lead to similar metabolic impairment in terms of processing and using endocytosed cholesterol, leading to excess storage in extraneural tissues. Diseased cells tend to rely on cholesterol trafficking pathways, while neurons mainly accumulate GM2 and GM3 gangliosides instead of cholesterol. Unfortunately, NPC1 and NPC2 proteins remain poorly understood. Neuropathological features may include neuronal storage, significant loss (especially of Purkinje cells), ectopic dendrites, neuroaxonal dystrophy, and similar changes that mirror AD. Clinical and biochemical features do not distinguish between those carrying NPC1 mutations (the dominant group) and those carrying rarer NPC2 mutations (rare group), though both groups can be present. Furthermore, this disease affects people of all ethnicities, with an estimated incidence rate estimated at 1 in 100,000 live births [148].

Changed cholesterol efflux from lysosomes disrupts neuronal firing patterns. The mechanism involves the upregulation of ABCA1 transporter, which decreases plasma membrane PtdIns(4,5)P2 content via PtdIns(4,5)P2-floppase activity, leading to decreased PtdIns(4,5)P2-floppase activity and therefore decreasing voltage-gated KCNQ2/3 potassium channel activity that regulates neuronal excitability when neurons affected by NPC1 disease are present, resulting in hyperexcitability. Lysosomal cholesterol efflux plays an essential role in shaping electrical and functional characteristics of neuronal plasma membranes under healthy as well as disease conditions [145] Figure 2.

NPC1 disease cases typically result from mutations of the NPC1 gene, coding for an integral membrane protein called NPC1. NPC1 protein resides in late endosomal vesicles within cells and plays an essential role in maintaining cholesterol and sphingolipid balance within them. Unlike enzyme deficiencies found in traditional lysosomal storage diseases (LSDs), however, NPC1 protein does not secrete itself from cells, making cross-correction unlikely through transduced cells alone; therefore, to achieve clinical benefits over CNS storage diseases with enzyme deficiencies alone, technical challenges would need to be met in order to achieve clinical benefits in comparison to CNS storage diseases caused by enzyme deficiencies alone [149]. 

Cells containing NPC2 mutations may be cross-corrected using an NPC2-enriched medium, making these patients suitable candidates for gene therapy or gene product transduction. Unfortunately, this approach does not address cholesterol accumulation caused by cells carrying NPC1 mutations [150].

An investigation of the role of GM2 ganglioside in NPC disease was undertaken by crossbreeding NPC mice with those lacking the ability to synthesize GM2 and GD2 gangliosides. Ganglioside-deficient mice demonstrated no neuronal GM2 ganglioside, supporting its role in NPC pathogenesis [149].

N-butyldeoxynojirimycin (NB-DNJ), an inhibitor of GSL biosynthesis, provided another approach to studying their roles in NPC mouse models by decreasing levels of all GlcCer-derived GSLs; substrate reduction therapy has proven useful against other glycosphingolipid storage diseases resulting from lysosomal enzyme deficiencies [151].

Erratic expression and distribution of TDP-43 were observed in an NPC mouse model as well as an in vitro human NPC neuronal model system. Cytoplasmic TDP43 sequestered in stress granules could indicate mutations, disease processes, or environmental stress, leading to neurodegeneration. Furthermore, an NPC neuronal model displayed abnormal gene expression under TDP-43 control at the RNA processing level, which suggests possible connections with the observed neuropathology of NPC [152]. 

NPCD, like other lysosomal storage disorders, is characterized by an accumulation of various lipids such as di- and triacylglycerols, phosphoinositides, sphingosine, sphingomyelin, GlcCer, and gangliosides. This accumulation may be affected by factors like increased pH causing decreased acid lipase activity, favorable physical/chemical interactions between stored lipids, and reduced enzyme activity due to stored lipids [153].

Surprisingly, although NPCD cells accumulate sphingomyelin, mice with the lysosomal hydrolase of this lipid knocked out (mimicking Niemann–Pick disease type A) do not accumulate cholesterol at the whole cell level. This observation points out the need for subcellular lipidomics to accurately quantify lipid accumulation due to anisotropic distribution and the potential significance of minor populations of lipids. Furthermore, suborganellar lipidomics may help shed more insight into this disease; specifically, by identifying whether cholesterol accumulates within intralysosomal membranes or on its limits, or both [153].

Despite the disease’s association with cholesterol trafficking anomalies, drugs aiming to lower cholesterol have not demonstrated effectiveness in altering the disease trajectory [154].

Arimoclomol, a derivative of bimoclomol, is known to bolster heat shock protein (HSP) gene expression. It aids in triggering HSPs, thus enhancing the inherent cellular protective mechanisms during cellular distress situations [155]. 

Treatment strategies for NPC have concentrated on several areas [156] (Table 8):-Lessening the volume of intra-lysosomal free cholesterol;-Curbing the production of glucosylceramide by inhibiting its synthase;-Regulating inflammatory reactions and immune responses;-Augmenting the transfer of free cholesterol from the lysosomal section into the cytosol; -Modulating the expression of genes vital for initiating cell differentiation by hindering histone deacetylases (HDAC);-Employing pharmacological chaperones to promote cellular protein repair pathways via the activation of molecular chaperones, like heat shock proteins;-Exploring the potential of gene therapy.

## 9. Tay–Sachs Disease

Tay–Sachs disease is an autosomal recessive condition caused by an enzyme deficiency called beta-hexosaminidase A, an important lysosomal enzyme responsible for breaking down GM2 ganglioside. Without enough of it being broken down by nerve cell lysosomes, an accumulation of this GM2 ganglioside occurs, which leads to nerve cell dysfunction. This condition falls under GM2 gangliosides, a group of lysosomal storage disorders where various defective peptides (alpha and beta subunits of beta-hexosaminidase A and the GM2 activator protein) contribute to the degradation of GM2 ganglioside. Infantile Tay–Sachs disease, also known as type 1 GM2 gangliosidosis or infantile Tay–Sachs disease, is typically characterized by almost-zero beta-hexosaminidase A activity; however, juvenile and late-onset forms exist with residual enzyme activity contributing to milder presentations of disease [158].

Dr. Warren Tay, a British ophthalmologist and physician, presented the case of an infant suffering from progressive weakness and changes to her eye’s yellow spot at one year of age before reporting similar cases within her family. Meanwhile, an American physician named Bernard Sachs independently described clinical manifestations and pathological features associated with infantile Tay–Sachs disease despite Dr. Tay’s reports. Since Sachs first identified autosomal recessive inheritance and increased Ashkenazic Jewish child occurrence in his initial observations of Tay–Sachs disease, further research confirmed this pattern and increased incidence. Robinson and Stirling identified two hexosaminidase isoenzymes A and B in the human spleen, which allowed Okada and O’Brien in California as well as Sandhoff in Germany to observe an absence of hexosaminidase-A activity among frozen tissue samples from patients suffering from Tay–Sachs disease [159].

Tay–Sachs disease is an autosomal recessive genetic condition caused by an insufficient supply of beta-hexosaminidase A (Hex A). This enzyme has two polypeptide chains encoded on chromosomes 15 and 5, known as alpha and beta subunits, respectively. These subunits combine to form an active enzyme responsible for breaking down specific substrates found within lysosomes, such as GM2 ganglioside. Hydrolysis of this substrate requires activator protein activation within its specific environment in the lysosome. Additionally, beta subunits can dimerize to form beta-hexosaminidase B (Hex B), an active enzyme that acts on water-soluble neutral substrates. Mutations in this beta subunit lead to Sandhoff’s disease, with deficient amounts of both Hex A and Hex B being present in its composition [160]. 

Mutations in the alpha-chain gene disrupt its catalytic function and consequently lead to the accumulation of GM2 gangliosides in neuronal tissue, leading to neurodegeneration and mental and motor retardation. Tay–Sachs disease’s clinical course depends on how much residual Hex A activity remains due to mutation severity. Mutations that completely disrupt Hex A activity cause the severe infantile form of Tay–Sachs disease, which presents early and rapidly, leading to death by early childhood. By contrast, the late infantile form presents later and has a later age of demise than its infantile form counterpart, with certain mutations associated with this variant of Tay–Sachs disease [160].

Adult Tay–Sachs disease is distinguished by its clinical variability, typically appearing during the second or third decade of life. Usually, there is less motor and neurological deterioration compared to other variants; psychosis often appears first. Missense mutations are usually responsible for this form of Tay–Sachs disease in adult individuals; Gly269Ser and 805G->A mutations found frequently within exon 7 are typically linked with it [161]. 

Tay–Sachs disease (TSD) was initially linked to Jewish families of Eastern European ancestry who resided within Sachs’ initial study area in New York City. TSD has now come to be recognized as not being limited to any one ethnic group but is widespread, impacting Blacks and Asians alike [162]. DNA sequencing technology has shown that TSD is caused by numerous mutations, disproving the myth that its causes were sole. French Canadians and Ashkenazi Jews, for instance, have shown evidence of genetic heterogeneity when it comes to TSD within these populations [163]. Furthermore, Ashkenazi Jews can have multiple mutations leading to TSD while Moroccan Jews with TSD can display distinct mutations as causes [164].

TSD and other severe forms of GM2 gangliosidosis, including its variants, are associated with an almost total deficiency in Hex A activity; however, total Hex activity remains within normal levels due to active Hex B production. On the other hand, in Sandhoff disease (SD), another form of GM2 gangliosidosis, only a minimal amount (2–4% of normal) of Hex activity can be detected despite normal levels of a-subunit mRNA. This is likely attributable to the low affinity between a-subunits, leading to detectable but limited amounts of Hex S. The less severe juvenile and adult-onset forms of TSD can often be traced back to mutations that allow some preservation of Hex A function (approximately 1–5% of normal) due to improper folding or dimerization of enzymes. Most missense mutations result in misfolded proteins, which are detected by the endoplasmic reticulum quality control system and degraded through its endoplasmic reticulum-associated degradation pathway (ERAD) before finally being processed through proteasome degradation [165]. 

Enzyme-Enhancing Therapy: When treating LSDs such as GM2 gangliosidosis or other lysosomal storage diseases, missense mutations often destabilize native folded proteins before reaching the lysosome for degradation. To address this, researchers have explored using pharmacological chaperones, or molecules that bind and stabilize enzymes and prevent their degradation; commonly employed are enzyme-specific competitive inhibitors. Another approach involves manipulating the endoplasmic Reticulum-Associated Degradation (ERAD) system to block the recognition of defective proteins or improve endogenous chaperone functions [166].

Gene Therapy: Neuropathic LSDs are an ideal candidate for gene therapy as their root causes lie within single genes with naturally secreted products that require corrective enzyme to restore deficient tissues; transducing just a small number of cells can deliver sufficient corrective enzyme to make up for deficiencies across an extensive region. Studies have demonstrated that even modest levels (around 10% of normal activity) in healthy individuals with pseudo-deficiencies are sufficient to avoid disease progression [167].

Enzyme Replacement Therapy and Cell Transplantation: This strategy has proven successful for some LSDs with systemic manifestations, including MPS I, Fabry disease, Pompe disease, MPS VI, and MPS II. Although enzyme therapy is currently being investigated in clinical trials for certain neuropathic LSDs, it may not provide an effective solution for GM2 gangliosidosis. B-hexosaminidase has a lower abundance in cultured microglia when compared with other lysosomal glycosidases, making it less amenable to enzyme replacement therapy [166]. Bone marrow transplantation has shown some promise for treating certain LSDs, such as human a-mannosidosis, but its effectiveness varies between diseases; for instance, in MPS IIIA (Sanfilippo type A), this therapy can extend life without providing neurological improvements [168]. 

Substrate Reduction Therapy: Miglustat, an iminosugar known as N-butyldeoxynojirimycin, belongs to a class of molecules known as glucosylceramide synthase inhibitors, which block glycolipid biosynthesis. While Miglustat showed promising results when tested on SD mice by improving lifespan and clinical features, its efficacy endpoint target in late-onset TSD patients was not satisfactory. Regardless, research into inhibitors to reduce substrate accumulation remains an active area [167,168].

One such success in this field is Cerdelga eliglustat, approved as Cerdelga, which has become a first-line therapy for type 1 Gaucher disease. Substrate reduction therapy using orally active small molecules such as Eliglustat has demonstrated therapeutic advantages and efficacy comparable to enzyme therapy for non-neuronopathic Gaucher disease patients. In an initial clinical trial involving patients with late-onset GM2 gangliosidosis, oral administration of pyrimethamine (PYR) at a lower dosage than typically used to treat parasitic diseases showed significant enhancement of Hex A activity in peripheral blood leukocytes; the degree of this enhancement correlated to plasma PYR concentration. The results also confirmed this improvement across Sandhoff and Tay–Sachs variants of GM2 gangliosidosis patients alike. Notably, PYR treatment’s increase of Hex A activity was limited to mutant forms of Hex A and not due to any non-specific lysosomotropic effect, as indicated by levels of Glcr and ss-galactosidase activity in plasma and leukocytes, which remained unchanged following PYR administration [169].

For Tay–Sachs disease, current data support a cellular pathology model whereby the alpha subunit (a) of an enzyme is translocated to the endoplasmic reticulum (ER), where it undergoes cotranslational glycosylation and folding via the calreticulin/calnexin cycle. When properly folded, this dimer forms with beta subunits (b). From there it travels to the Golgi apparatus, where mannose-6-phosphate signals direct it back toward lysosomes for its intended function [170] Figure 3.

However, in cases where ER folding of the alpha subunit becomes impaired due to mutations like G269S and E482K, its glycan trimming process—often in conjunction with OS-9 association—can initiate a disposal pathway. When this occurs, retrotranslocon protein channels allow access to the cytosol. PNGase removes any remaining glycans before degradation occurs at 26S proteasome and ultimately results in the removal of improperly folded enzyme molecules, helping explain the pathology of Tay–Sachs disease [171]. 

## 10. Fahr’s Syndrome

Basal ganglia calcification, also referred to as Fahr’s syndrome or Fahr’s disease, is an extremely rare neurological condition with an incidence rate of less than 1 out of 1 million individuals, usually affecting those in their third and fourth decades of life [170]. First described by German neurologist Karl Theodor Fahr in 1930, Fahr’s disease manifests itself by abnormal calcium deposition within certain brain regions responsible for controlling movements, such as basal ganglia, thalamus, dentate nucleus, cerebral cortex, cerebellum, and hippocampus [172].

Patients often present with extrapyramidal symptoms. Furthermore, they may also exhibit cerebellar dysfunction, speech difficulties, dementia, and neuropsychiatric signs and symptoms. Genetic studies have implicated various loci in Fahr’s disease development. One commonly implicated locus (IBGC1) lies at 14q, while two other loci have been discovered on chromosome 8 and two more on 2. Additionally, loss-of-function mutations found within SLC20A2 on chromosome 8 have been suggested as contributing to its pathophysiology [173]. 

As part of an appropriate diagnostic approach for Fahr’s syndrome, it is advisable to conduct sequencing of the SLC20A2 gene first. If no mutations or deletions are identified within SLC20A2, deletion/duplication analysis could then be explored further. In the absence of either identifiable mutations or deletions within SLC20A2, sequence analysis on PDGFRB would then be necessary; should no disease-causing mutations or deletions be identified through molecular genetic tests, then other genetic disorders linked with brain calcifications should be explored further [173].

Fahr’s syndrome may arise due to various underlying conditions, including endocrine disorders (such as idiopathic hypoparathyroidism, secondary hypoparathyroidism, and hyperparathyroidism), adult-onset neurodegenerative conditions (e.g., neurodegeneration with brain iron accumulation disease), infectious diseases such as intrauterine or perinatal infections, as well as inherited syndromes like Aicardi–Goutieres syndrome or tuberous sclerosis complex [174].

Seizures and movement disorders related to Fahr’s syndrome may be improved by correcting phosphate and calcium levels. Treatment with alpha hydroxy vitamin D3 and corticosteroids has also been shown to reverse neurological deficits in some instances [174]. Clonazepam and atypical antipsychotics may also help with managing symptoms for those living with Fahr’s syndrome, though lithium use should be used with caution as it may increase seizure risk. Furthermore, carbamazepine, benzodiazepines, and barbiturates may worsen underlying gait disorders; therefore, these strategies should also be employed carefully when treating patients living with Fahr’s syndrome [175].

Genes for Basal Ganglia Calcification:

Mutations of the SLC20A2 gene found on 8p11.21 can be passed down autosomally dominantly. Proper inorganic phosphate transport is essential in keeping the calcium and phosphate balance within cells intact, with impaired PiT2 function leading to the accumulation of calcium phosphate deposits in extracellular matrix tissues of vessels. Research using Slc20a2-knockout mice (KOs) has shed light on pathophysiology associated with SLC20A2, showing increased cerebrospinal fluid, suggesting potential therapeutic implications as a therapeutic option.

Located on 1q25.3, the XPR1 gene is tightly associated with PiT2. It encodes for the xenotropic and polytropic retrovirus receptor that plays an essential role in exporting phosphate from cells, contributing directly to intracellular phosphate homeostasis as well as calcium deposition in some instances. In families, mutations of this gene may be passed down as autosomal dominant traits [176].

PDGFRB is a gene linked to familial primary brain calcification (PFBC). It encodes one of two receptors for platelet-derived growth factor (PDGF), with subunit ss being its main ligand and playing an essential role in maintaining the integrity of the blood–brain barrier. Loss-of-function mutations could disrupt pericyte permeability surrounding brain blood vessels, resulting in the deposition of calcium deposits within brain tissues. Mutations in both PDGFB and PDGFRB genes exhibit autosomal dominant inheritance patterns. PDGFB plays an essential role in pericyte recruitment, blood–brain barrier regulation, angiogenesis, and calcium regulation through BBB disruption, leading to progressive calcinosis and disruption. Furthermore, PDGF proteins may regulate phosphate transporters such as XPR1 and PiT in the brain [177].

PTH resistance disorders may play a part in some cases of brain calcification. When that is the case, loss of function of GNAS on the maternal allele can cause basal ganglia calcification—although not considered primary familial brain calcification (PFBC). GNAS is a complex imprinted locus responsible for producing multiple transcripts via alternative splicing and promoters [178].

Keller et al. conducted genetic analysis on six families to discover that mutations in the gene encoding PDGF-B (located at 22q13.1) could be responsible for familial IBGC in patients who previously tested negative for SLC20A2 and PDGFRB mutations. Furthermore, these experiments also showed that mice that expressed 50% less endothelial PDGF-B developed significant brain calcification within one year [179].

However, it should be remembered that mutations of PDGF-B and SLC20A2 each have different pathophysiological mechanisms. Loss of endothelial PDGF-B has been associated with pericyte deficiency as well as blood–brain barrier deficiency; on the other hand, loss of SLC20A2, as a member of the type III sodium-dependent phosphate transporters family, has been linked with accumulations of inorganic phosphate in the brain, leading to calcium phosphate deposition [180]. 

To date, multiple treatment modalities have been explored in Fahr’s patients, striving for remission or stabilization. These treatments span a range of biological hypotheses and derive from limited clinical observations. Medications are primarily prescribed to manage symptoms like anxiety, depression, obsessive–compulsive disorder, and dystonia. For urinary incontinence, oxybutynin is preferred, while antiepileptics are chosen for seizures. Interestingly, seizures and movement disorders associated with Fahr’s syndrome, especially those linked to parathyroid imbalances, can be addressed by regulating phosphate and calcium levels. As an illustration, employing alpha-hydroxy vitamin D3 combined with corticosteroids has reversed some neurological deficits [181] Table 9.

Research has pointed toward various SLC20A2 mutations as potential culprits behind Fahr’s disease, which is a subset of primary familial brain calcification (PFBC). Yet, there have been only a handful of patient-derived induced pluripotent stem cell (iPSC) models to study this condition. Notably, Zhang and colleagues discovered a novel SLC20A2 mutation in a Fahr’s disease-affected family and managed to secure dermal fibroblasts from a member of this family. They successfully reprogrammed these fibroblasts into iPSCs using episomal plasmids carrying genes like OCT3/4, SOX2, KLF4, LIN28, and L-MYC. Such efforts present valuable resources and platforms, paving the way for more detailed investigations into Fahr’s disease mechanisms. This could potentially enhance the development and assessment of both drug and gene-based interventions [182] Table 9.

## 11. Conclusions and Future Perspectives 

Neurodegenerative ailments have grave implications, and conventional medications often fall short. Presently, stem cell therapy is gaining traction as a potentially transformative solution, but much of the current research is rooted in animal studies, creating a knowledge void regarding its long-term impacts and results in humans. Positive outcomes in animals demand more rigorous investigation before being translated into human therapies. Among the various neurodegenerative conditions, Parkinson’s disease and amyotrophic lateral sclerosis have received more attention compared to Huntington’s disease and Alzheimer’s disease. Before stem cell therapies can become a mainstream treatment for these conditions, numerous hurdles, such as cost, safety concerns, the expertise involved, and post-procedure monitoring, need to be addressed [183].

Traditional treatments for neurodegenerative diseases, such as small-molecule medications and immunotherapy aimed at harmful proteins, largely alleviate symptoms without actually stopping or reversing the disease’s advance. New therapeutic techniques like autophagy therapy, miRNA therapy, and stem cell therapy have been introduced. However, they face challenges such as difficulty in crossing the blood–brain barrier, unexpected side effects, and challenges in addressing intracellular proteins. The emergence of protein-targeted degradation technologies brings hope by potentially targeting proteins previously considered unreachable for drug interventions. These novel degradation methods each have their distinctive mechanisms and objectives, accompanied by their advantages and drawbacks. This examination explores their present roles in neurodegenerative diseases, weighs their strengths and weaknesses, and foresees their future trajectory in the domain [184].

Gene therapy offers a hopeful avenue for addressing neurodegenerative diseases. Transitioning these therapies from theoretical frameworks to practical clinical applications has, however, been difficult. Preliminary studies indicate that delivering gene therapies to the central nervous system is safe and typically well-tolerated. To improve delivery, new vectors, such as AAV9, are being researched. AAV9 stands out for its ability to cross the blood–brain barrier and target neurons predominantly. Yet, its efficiency might fluctuate based on factors like the recipient’s age at the time of treatment. Also, the high production costs of AAV9 for human testing make its broad application challenging. Another exciting advancement in this domain is the tricistronic lentiviral vector, highlighted in the Prosavin trial. Efforts are also ongoing to refine gene therapy delivery methods to the CNS, including intracerebroventricular, intrathecal, and direct brain and spinal cord injections. However, these delivery advancements would be moot if the genes delivered are not efficacious. Therefore, a key priority is the discovery and assessment of new therapeutic genes, underpinned by a richer understanding of the origins and evolution of neurodegenerative conditions. As our knowledge deepens, we expect improved early diagnosis, facilitating interventions before significant cellular damage [185].

## Figures and Tables

**Figure 1 ijms-24-13006-f001:**
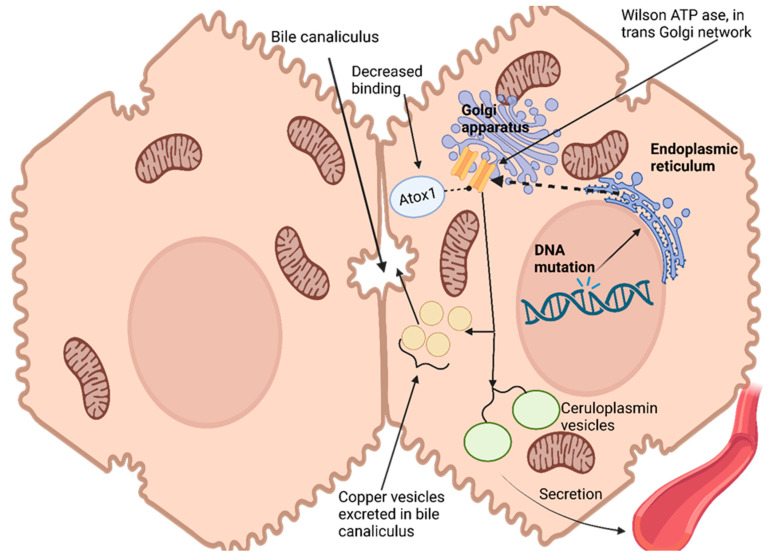
The proposed model depicts various potential molecular mechanisms responsible for the disturbed copper homeostasis observed in hepatocytes expressing Wilson’s disease ATPase mutations. Under normal circumstances, this ATPase normally resides in the trans-Golgi network, but when exposed to high copper levels, it relocates into a vesicular compartment of its cell’s cytoplasm, where it accumulates copper before returning to its trans-Golgi network and eventually excreting copper into its bile.

**Figure 2 ijms-24-13006-f002:**
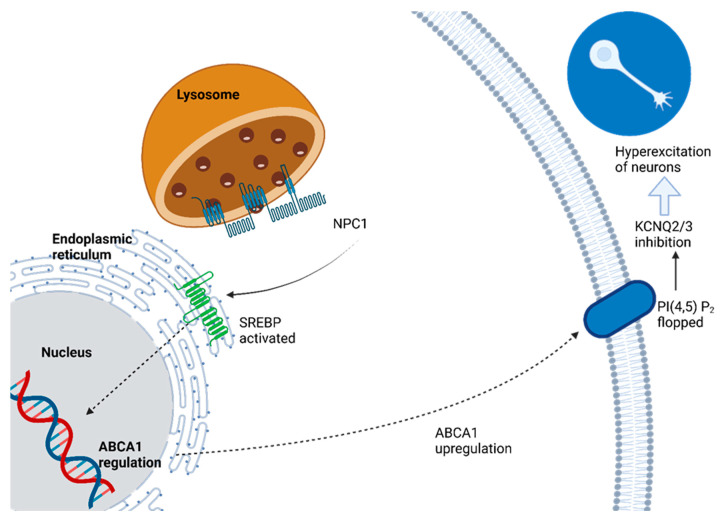
NPC1 disease is a neurodegenerative disorder caused by genetic mutations of the lysosomal cholesterol transporter NPC1. Cholesterol accumulates in these organelles and begins its pathological process, ultimately resulting in greater excitability of neurons affected.

**Figure 3 ijms-24-13006-f003:**
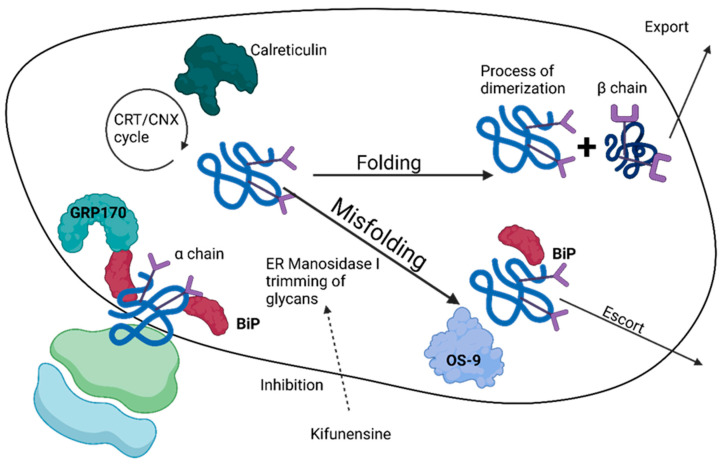
This diagram depicts the complex cellular model of Tay–Sachs disease. After alpha subunit undergoes glycosylation into the ER and structural stabilization from calreticulin/calnexin cycle, it can properly fold, forming the beta subunit, and both will be exported to the Golgi. If alpha subunit undergoes improper folding, glycans are removed before proteasomal degradation. A possibility to avoid this issue is Kifunensine, an inhibitor for ER Manosidase I trimming of glycans.

**Table 1 ijms-24-13006-t001:** Overview of targeted molecular therapies in frontotemporal dementia.

Study Focus	Key Findings and Techniques	Citation
Origin and employment of iPSCs	Derived from somatic cells like fibroblasts. Uses agents Oct4, Klf4, Sox2, and c-Myc.	[22]
iPSC models for FTLD-tau	Focuses on fibroblasts from MAPT alterations.	[25]
Patient-sourced iPSCs with 10 + 16 mutation	Noted surge in 4R tau expression, leading to increased 4R:3R tau isoform ratio.	[26]
Tau pathology in N279K iPSC-derived neurons	Observed shifts in 4R:3R tau isoform balance, with increased 4R tau levels.	[27]
iPSCs from patients with N279K mutation	Differentiated into NPCs. Used CRISPR/Cas9 for isogenic controls. Differentiated into astrocytes using Sox10 and growth factors.	[27]

**Table 2 ijms-24-13006-t002:** Molecular pathways useful in spinocerebellar ataxias treatment.

Molecular Target	Pathway	SCA Subtype	Possible Medication
PP2	PP2	1, 12	PP2-mediated regulators
PRKC	PRCK	1, 14	PRKC-mediated regulators
Gene transcriptors	Multiple	1–3, 6, 7	HDACis
Aggregation	Autophagy Transglutaminase	1–3, 6, 7, 17, DRLPA	Rapamycin Cystamine
Chaperons	HSR, UPR	1–3, 6, 7, 17, DRLPA	Arimoclomol
Ubiquitin	UPS	1–3, 6, 7, 17, DRLPA	UPS derivates
Mitochondrial approach	Multiple	Any	Coenzyme Q10
Calcium activity	Calcium mechanisms	1, 6	Ca^2+^ blockers
Dopamine pathway	Dopamine	1–3, 6, 17, 27	Levodopa, anticholinergic, and dopaminergic pharmaceutical therapies
Neurotransmitters	GABA	Any	Glutamate inhibitors
Ataxins	RNAi	Any	RNAi
Caspases	Caspases	Any	Cystamine

PP2—serine/threonine protein phosphatase 2; PRKC—protein kinase C; HDACis—histone deacetylase inhibitors; DRPLA—dentatorubral pallidoluysian atrophy; HSR—heat shock response; UPR—unfolded protein response; UPS—ubiquitin/proteasome system; GABA—gamma aminobutyric acid; RNAi—RNA interference.

**Table 3 ijms-24-13006-t003:** Overview of targeted molecular therapies in spinocerebellar ataxias.

Treatment/Research Focus	Research Focus	Citation
AAV vector method targeting ATXN1	Introducing AAV vectors with shRNA targeting ATXN1 led to reduced mutant ATXN1 expression. This improved motor function and normalized Purkinje cells. Research now focuses on specifically targeting mutant alleles in polyQ disorders, including SCA3.	[42,43]
Research on polyQ expansion in SCA17	PolyQ expansion in TBP affects its DNA binding and interactions with transcription regulators. This might reduce the expression of specific genes, impacting neuronal function. However, the majority of genes remain unaffected.	[44]

**Table 5 ijms-24-13006-t005:** Overview of targeted molecular therapies in Progressive Supranuclear Palsy.

Treatment/Drug	Key Details and Outcomes	Citation
Levodopa with carbidopa or benserazide	- Primary drug for dopaminergic replacement in PSP. - PSP-P subtype shows a better response. - 20–30% of pathologically confirmed and 20–40% of clinically diagnosed PSP patients have positive effects when combined with another dopaminergic drug like amantadine.	[96]
Donepezil	- Can exacerbate motor symptoms while improving cognitive abilities in PSP patients.	[97]
Amitriptyline	- Anecdotal claims of improvement in motor functions, but limited data on its effects on gait issues in PSP.	[98]
Amantadine	- NMDA-receptor antagonist with inconsistent results in PSP.	[98]
Carbidopa-levodopa	- Indicated as the most effective agent in a study of 147 patients, improving Parkinsonism in 20% of the treated group.	[98]

**Table 6 ijms-24-13006-t006:** Overview of targeted molecular therapies in corticobasal dementia.

Treatment/Drug	Application/Condition	Key Outcomes	Citation
Dopaminergic treatments, benzodiazepines, anticholinergics	Parkinsonism	Typically ineffective and might lead to side effects such as cognitive decline.	[98]
Levodopa	Parkinsonism in CBD/CBS	Minimal effects, with some reports indicating brief mild to moderate improvement.	[101]
Botulinum toxin injections	Dystonia in CBS	Effective for pain relief, enhancing hygiene, countering secondary contractures, and occasionally improving limb functionality in early disease stages.	[108]
Benzodiazepines (e.g., Clonazepam), anticholinergic drugs, muscle relaxants	Dystonia	Oral medications like these are often tried but seldom prove effective.	[109]
Clonazepam	Myoclonus	Shows a favorable response, especially effective.	[109]
Levetiracetam, gabapentin, valproic acid	Myoclonus	Can be beneficial, with some accounts of use.	[109]

**Table 7 ijms-24-13006-t007:** Overview of targeted molecular therapies in Wilson’s disease.

Model/Component	Description/Effect	Citation
R778L mutation model	Differentiated into hepatocyte-like cells (HLCs)	[143]
HLCs with R778L mutation	Showed heightened susceptibility to excessive copper, more sensitive to copper-induced cytotoxicity	[143]

**Table 8 ijms-24-13006-t008:** Overview of targeted molecular therapies in Niemann–Pick disease.

Treatment/Therapeutical Approaches	Description/Effect	Citations
Cholesterol-lowering drugs	Not effective in altering disease trajectory	[154]
Regulating inflammatory reactions		[154]
Curbing glucosylceramide production	By inhibiting its synthase	[155]
Augmenting free cholesterol transfer	Stimulating the transfer from lysosomal section into the cytosol	[155]
Modulating gene expression	By hindering histone deacetylases (HDAC)	[155]
Pharmacological chaperones	Promote cellular protein repair pathways via activation of molecular chaperones like heat shock proteins	[155]
HP-β-CD	Cyclic oligosaccharide; limited by inability to penetrate blood–brain barrier	[154]
Arimoclomol	Enhances heat shock protein gene expression, aiding cellular protective mechanisms	[155]
Miglustat	Sole officially approved treatment in EU for neurological symptoms; may slow or mitigate disease progression	[157]

**Table 9 ijms-24-13006-t009:** Overview of targeted molecular therapies in Fahr’s syndrome.

General Medications	Manage Symptoms: Anxiety, Depression, OCD, Dystonia	Citations
Oxybutynin	Treat urinary incontinence	[181]
Antiepileptics	Address seizures	
Phosphate and calcium level regulation	Address seizures and movement disorders, especially those linked to parathyroid imbalances	
Alpha hydroxy vitamin D3 + corticosteroids	Reversed some neurological deficits	
Clonazepam and atypical antipsychotics	Provide therapeutic benefits	
Lithium	Prescribed with caution due to increased seizure risk	
Carbamazepine, benzipenes, barbiturates	Might intensify gait disturbances	
Antidepressants and anxiolytics	Prescribed with caution due to potential side effects at lower thresholds in Fahr’s syndrome patients	
iPSC models (SLC20A2 mutations)	Research tool for studying Fahr’s disease mechanisms, potential for drug- and gene-based intervention studies	[182]

## Data Availability

All data is available online on libraries such as PubMed.

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
