# Peer review of "Decoding Neurodegeneration: A Comprehensive Review of Molecular Mechanisms, Genetic Influences, and Therapeutic Innovations"

_ijms, 2023, doi:10.3390/ijms241613006_

Round 1

Reviewer 1 Report

Overall, great primer on the clinical subtypes of various neurodegenerative disorders in addition to treatments and studies available. Authors provide adequate summary of studies, with a somewhat historical perspective. What is missing is more references about clinical and preclinical treatments as well as cell culture studies. There are a lot of paragraphs (e.g. those starting in line 149, 322, 331, etc) that are listing a lot of findings but missing the primary literature references associated with them. 

some other minor grammar changes, for instance Line 18: it's pathologies' and not pathology's; Line 149, the word after the colon is not capitalized and that remains afterwards.

Author Response

Dear Reviewer,

We have added more clinical and preclinical studies focused on the treatment of neurodegenerative disorders, as well as cell culture studies, if they aren’t already discussed in the body of subchapter, those newly included parts are listed at the end of each subchapter

Regarding paragraphs whom need the primary literature references, the paragraph starting with line 322 lists the findings from only the article mentioned. For line 149 and 331, I’ve added more references for a precise bibliography.

I’ve corrected the minor grammar mistakes, those mentioned as well as others identified.

Thank you for your helpful review!

Reviewer 2 Report

Reviewer Comments

In the present review article on “Decoding Neurodegeneration: A Comprehensive Review on Molecular Mechanisms, Genetic Influences, and Therapeutic Innovations”, the authors have discussed the molecular mechanisms and the implications regarding those pathology’s clinical advancement, and an analytical overview of genetic mutations in the case of neurodegenerative disorders.

The topic is interesting however the papers lack a final summary, a concluding remark, and a future prospective. The paper needs to be revised thoroughly.

Scientific comments

1.      In the abstract authors have mentioned the ‘genomic sequences techniques’ however in the main body of the article there is no further discussion regarding genomic sequences techniques.

2.      Diagrammatic representation is not up to the mark. Improve Figure 1.

3.      Line 604-621 have only 1 reference [89], Is almost 20 lines taken from a single paper? Add more references.

4.      In Figure 3 missing the legend.

5.      There is no concluding remark. Add a concluding remark with a future perspective.

6.      At least give some expert comments from your side regarding the therapeutic innovations for Neurodegenerative disease.

7.      The major importance of this article is the Molecular Mechanisms, Genetic Influences, and Therapeutic Innovations however in the text, these were vaguely described or completely skipped.

8.      Without a more detailed explanation of current researchers etc and an expert opinion of the importance of the results this review is mostly just information that could have been copied and pasted from other reviews already published. 

9.      Most of the paragraphs of 8-10 lines are taken from one paper only. More suitable references should be added.

Minor comments

10.  line 100-101; check these lines for repetitions.

11.  Check grammar for sentences 146-147.

12.  Line 666-672 add the reference from where this data is collected.

13.  Line 673-677, Sentence is not understandable. Split it into two.

14.  Check lines 677-679. rewrite the sentence.

15.  Sentences 692-699, are not easy to understand this large sentence. Rewrite it properly.

16.  Provide a suitable reference for lines 716-720.

17.  line 114-18 is a large sentence, divide it into two.

18.  Check the reference pattern and revised the reference accordingly.

Many grammatical mistakes and large sentences are unable to understand.

Author Response

Dear Reviewer,

We have eliminated the “genomic sequences techniques” from the abstract, our study didn’t focus on this subject

We have added more references or modified the improper ones, rephrased/rewritten the mentioned sentences, eliminated repetitions and grammar is now improved

We wrote a concise caption for Figure 3

More detailed explanations of current researches were added, especially regarding clinical treatments and possibilities of therapeutic management

Conclusions and Future Perspectives are properly listed at the end of the manuscript

I’ve added studies with clinical and preclinical treatments, as well as cell culture studies, with proper literature references

Expert comments were added in new paragraphs regarding therapeutic modalities and cell studies

Now the reference pattern is consisted, IEEE reference format

Round 2

Reviewer 2 Report

Reviewer Comments

In the present article “Decoding Neurodegeneration: A Comprehensive Review on Molecular Mechanisms, Genetic Influences, and Therapeutic Innovations,” the authors have discussed the Molecular Mechanisms, Genetic Influences, and Therapeutic Innovations for neurodegenerative disorders.  

In the revised MS, the authors have added the content and updated the references however the quality of the paper is not improved accordingly.

Comments

1.      Line 139, [Others molecular subtypes]:, check this highlighted part and do corrections if needed.

2.      Line 298-30, three paragraphs are from the same paper [29]. Why you have cited the same reference three times continuously?

3.      Full form & abbreviated form in brackets for Corticobasal degeneration (CBD), Alzheimer's disease (AD), progressive supranuclear palsy (PSP), and DLB have been written many times in MS. Correct the repetition.

4.      Provide a table for a summary of molecular targeted therapies and their implementation approaches in neurodegenerative diseases discussed in your review.

5.      Concluding remarks and future prospective is more like a summary of your review paper. Write a proper concluding remark and future prospective.

6.      Check the reference pattern. Revised the reference according to the IJMS pattern.

Minor English editing is required.

Author Response

Dear Reviewer,

We truly appreciate your suggestions and comments!

Point-by-point response:

1) We’ve made the corrections for more concise explanations

2) Proper references were added for the mentioned paragraphs. In plus, more adequate citations were added in those paragraphs with only a few references.

3) We’ve made the modifications. When those terms are first mentioned into the manuscript, full form and abbreviated form are written. Further in the text, only the abbreviated form is written. Moreover, in each subchapter title, full form and abbreviated form are provided for a coherent structure of the manuscript.

4) New tables of molecular targeted therapies has been added, one for every chapter with treatment focus.

5) Future perspectives were added.

6) Reference pattern is now ACS, according to IJMS indications and guidelines.

Moreover, the English language was proofread and corrected.

Thank you for your most significantly helpful review!

Round 3

Reviewer 2 Report

Reviewer Comments

In the present article “Decoding Neurodegeneration: A Comprehensive Review on Molecular Mechanisms, Genetic Influences, and Therapeutic Innovations,” the authors have discussed the Molecular Mechanisms, Genetic Influences, and Therapeutic Innovations for neurodegenerative disorders.  

In the revised MS, the authors have added the content and updated the references as per suggestions.

Comment

1. Check the citation pattern all over the MS. Citation pattern is different in the text and in the table.

Author Response

Dear Reviewer,

Citation pattern is now consistent in both text and tables, according to ACS citation format

Thank you for you review and the helpful recommendations!